# Language-specific neural dynamics extend syntax into the time domain

Cas W. Coopmans [1,2,3]*, Helen de Hoop[3], Filiz Tezcan[1,2], Peter Hagoort[1,2], Andrea E. Martin[1,2]

1 Max Planck Institute for Psycholinguistics, Nijmegen, the Netherlands, 2 Donders Institute for Brain, Cognition, and Behaviour, Radboud University, Nijmegen, the Netherlands, 3 Centre for Language Studies, Radboud University, Nijmegen, the Netherlands

* cas.coopmans@mpi.nl

**Data Availability Statement:** All data and analysis files are available on the Radboud Data Repository: https://doi.org/10.34973/m1vp-hc15.

**Funding:** PH was supported by the Netherlands Organization for Scientific Research (NWO;

## Abstract

Studies of perception have long shown that the brain adds information to its sensory analysis of the physical environment. A touchstone example for humans is language use: to comprehend a physical signal like speech, the brain must add linguistic knowledge, including syntax. Yet, syntactic rules and representations are widely assumed to be atemporal (i.e., abstract and not bound by time), so they must be translated into time-varying signals for speech comprehension and production. Here, we test 3 different models of the temporal spell-out of syntactic structure against brain activity of people listening to Dutch stories: an integratory bottom-up parser, a predictive top-down parser, and a mildly predictive left-corner parser. These models build exactly the same structure but differ in when syntactic information is added by the brain—this difference is captured in the (temporal distribution of the) complexity metric "incremental node count." Using temporal response function models with both acoustic and information-theoretic control predictors, node counts were regressed against source-reconstructed delta-band activity acquired with magnetoencephalography. Neural dynamics in left frontal and temporal regions most strongly reflect node counts derived by the top-down method, which postulates syntax early in time, suggesting that predictive structure building is an important component of Dutch sentence comprehension. The absence of strong effects of the left-corner model further suggests that its mildly predictive strategy does not represent Dutch language comprehension well, in contrast to what has been found for English. Understanding when the brain projects its knowledge of syntax onto speech, and whether this is done in language-specific ways, will inform and constrain the development of mechanistic models of syntactic structure building in the brain.

## 1. Introduction

How the brain transforms continuous sensory stimulation into cognitive representations is a major question in human biology. Because sensory input alone is typically insufficient to uniquely determine a coherent and reliable percept, perception requires the brain to add information to its analysis of the physical environment [1]. Language processing is a prime

Gravitation Grant 024.001.006 to the Language in Interaction Consortium, https://www.nwo.nl/en/ researchprogrammes/gravitation). AEM was supported by an Independent Max Planck Research Group (https://www.mpg.de/max-planck-research-groups) and a Lise Meitner Research Group "Language and Computation in Neural Systems" from the Max Planck Society (https://www.mpg.de/804961/lise-meitner-groups), by NWO (grant 016.Vidi.188.029, https:// www.nwo.nl/en/calls/nwo-talent-programme), and by Big Question 5 (to RC & AEM) of the Language in Interaction Consortium funded by an NWO Gravitation Grant (024.001.006) awarded to PH. CWC was supported by an NWO grant (016. Vidi.188.029) awarded to AEM. The funders played no role in the study design, data collection and analysis, decision to publish, or preparation of the manuscript.

**Competing interests:** The authors have declared that no competing interests exist.

**Abbreviations:** AP, adjectival phrase; ATL, anterior temporal lobe; CWT, continuous wavelet transform; ERP, event-related potential; IFG, inferior frontal gyrus; mTRF, multivariate temporal response function; NP, noun phrase; PB, prosodic boundary; PTL, posterior temporal lobe; ROI, region of interest; VIF, variance inflation factor; VP, verb phrase.

example: one must know a language in order to perceive it from speech, so comprehending spoken language requires the encoding of information beyond what is presented in the raw speech signal. This includes the formation of syntactic structures—the "instructions" by which words are hierarchically combined for interpretation, as held by many theories of human language. What is noteworthy is that these theories assume that (the brain's) representations of syntactic rules and structures are abstract and not bound by time, yet they must be translated into temporal signals to comprehend and produce speech. This process, by which syntactic structure is incrementally built up in time, is termed "parsing" in psycholinguistics. Here, we compare 3 competing ways in which syntactic information can be metered out by the brain. These different models of parsing build exactly the same syntactic structure, but differ in the temporal dynamics of structure building. We thus investigate not only what kind of linguistic information the brain projects onto its perceptual analysis of speech, but also when it does so.

As this discussion makes clear, integrating the computational level of linguistic theory with the implementational level of cognitive neuroscience requires a linking hypothesis that specifies a connection between 2 fundamentally different types of data—atemporal linguistic structure and continuously varying neural dynamics [2–4]. When it comes to studying syntactic structure building in the human brain, it is therefore important to be explicit about the structure of the syntactic representations, the algorithmic procedures for computing these representations in real time, and the linking theory that maps the output of these algorithms onto neural signals [5–12]. One promising approach relies on neuro-computational language models, which are computationally precise models of language processing that define a measure of processing difficulty for every word in a data set that reflects everyday language use [13,14]. By determining whether this measure reliably predicts brain activity elicited by the words in that data set, we can establish whether there is evidence for the hypothesized linguistic computations in the neural signal.

In the current study, we compare different neuro-computational models in terms of their ability to predict brain activity of people listening to Dutch audiobook stories. In order to connect syntactic structure with neural dynamics, we adopt the complexity metric "incremental node count," which corresponds to the amount of structure that is assumed to be built when incorporating a word into the hierarchical structure of the sentence. Incremental node count is an appropriate linking hypothesis because it both directly reflects atemporal syntactic structure and also varies in time. By extending syntax into the time domain, the dynamics of incremental node count model the temporal distribution of perceptual cues for the inference of syntax (Fig 1). Here, we regress node count against brain activity in a time-resolved manner, in order to uncover the neural dynamics of syntactic structure building during natural story listening.

## 1.1. Neuro-computational models of sentence comprehension

Brennan [13] defines neuro-computational models as consisting of a parser $P_{G,A,O}$ that contains a grammar $G$ with rules to construct representations, an algorithm $A$ for incrementally applying the grammar word by word, and an oracle $O$ that resolves indeterminacies. When applied to a sequence of words $w_1, w_2, \ldots w_n$, $P_{G,A,O}$ yields a sequence of mental states $m_1, m_2, \ldots m_n$, which correspond to (partial) syntactic structures. These mental states can be quantified via an auxiliary hypothesis or linking rule, often referred to as complexity metric in psycholinguistics because of the way in which it quantifies language processing complexity [15]. The complexity metric $C$ thus represents the magnitude of the neural state in quantifiable cognitive terms; it stands for estimated brain states. The estimated brain states are linked to the observable neural signal via a response function $R$.

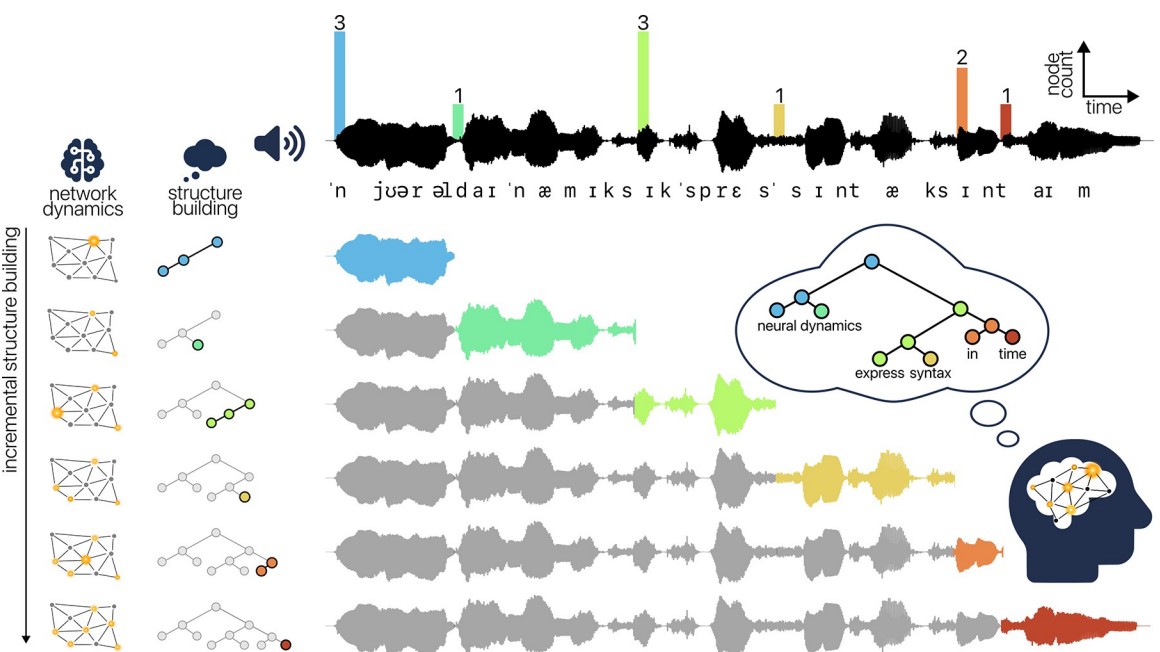

**Fig 1. Neural dynamics of incremental structure building in spoken language comprehension.** Speech is segmented into words, which are incrementally integrated into a hierarchical syntactic structure with an interpretation. The amount of structure built at each word, quantified via incremental node count, is reflected in the dynamics of the network, which lead to a neural signal measurable with magnetoencephalography.

This neuro-computational method has the potential to evaluate the cognitive and neural realization of linguistic constructs to the extent that there are appropriate linking hypotheses about how these constructs are manifested in neural activity. One assumption underlying this approach is that a model that captures knowledge of linguistic structure gives rises to measures that are predictive of experimental data. It is promising that the results of recent neuroimaging studies point in this direction. For example, grammars that compute hierarchical structure account for variance in neural activity over and above the variance accounted for by sequence-based models [10,16–20], and hierarchical grammars that naturally represent long-distance dependencies, which are ubiquitous in natural languages, uniquely predict activity in brain areas commonly linked to syntactic structure building [21–24]. These findings reinforce the view that grammars that are well equipped to account for the types of structures found in natural language are also required to adequately model the activity of the brain when it incrementally computes these structures.

**1.1.1. Three models of the temporal spell-out of syntactic structure.** Parsing models specify how syntactic parse states incrementally unfold in time during language comprehension. In the current work, these parse states correspond to partial syntactic structures, which are generated via the rules of X-bar theory (the grammar $G$; [25,26]). We chose to construct X-bar tree structures because they are appropriately expressive to deal with natural language structures, which is to say that they can capture the different types of hierarchical dependencies found in natural languages (e.g., long-distance dependencies, displacement, see Section 2.3). Recent work using neuro-computational models has shown that complexity metrics derived from these and similar kinds of structures account for brain activity above and beyond activity predicted by both sequential models and models that are hierarchical but less expressive [21–24]. We further assume a perfect oracle $O$, which resolves temporary ambiguities in the right

way [16,27,28]. This means that the parser builds the correct structure at any point in the parse, even when faced with locally ambiguous input.

Any given syntactic structure may be built in different ways, depending on the parsing algorithm *A* that is adopted. We will be concerned with 3 algorithms for building structure: a top-down algorithm, a bottom-up algorithm, and a left-corner algorithm (see [29]). The top-down parsing method works via expansion of rewrite rules. Before each word, all rules necessary to attach the upcoming word to the structure are expanded (e.g., in the rewrite rule VP ➜ V NP, the VP is expanded as V and NP). As these rules are applied based on the left-hand side of the rule and in advance of each word, this method builds constituent structure entirely predictively. The bottom-up parsing method, instead, builds constituent structure in a non-predictive manner, as it postulates a constituent node only after all of its daughter nodes are available. This process is referred to as reduction: when all information on the right-hand side of a rule is available, the input is reduced to the constituent node (e.g., in VP ➜ V NP, the input V and NP is reduced to VP). In between these 2 parsing methods is a mildly predictive left-corner strategy, which works via projection. A constituent node is projected after the very first symbol on the right-hand side of the rewrite rule (its left corner) is seen (e.g., the VP node is projected when V is available, but before NP is seen). This strategy is only mildly predictive because, while it requires input to build structure (in contrast to the top-down strategy), the input can be incomplete (in contrast to the bottom-up strategy).

To illustrate how these strategies differ in temporal spell-out, consider a simple sentence like "the boy sleeps." The total number of operations (expand, reduce, project) for the 3 parsing methods is the same, but the time points at which they are applied differ. The predictive top-down parsing strategy postulates structure early in time. For example, 3 operations are applied before "the," corresponding to expansion of the S, NP, and D nodes (Table 1). Only 1 operation is applied at "the" on the bottom-up method, because there is complete evidence for the determiner node D only. The next word "boy" is the second word of the noun phrase (NP), so 2 operations are applied bottom-up, but only one is applied top-down. What this simple structure illustrates is that these parsing methods differ in the dynamics of structure building. They make different claims about the time points at which processing complexity is high, and these differences will only be magnified when the sentences become longer and more complex.

Here, we represent the number of parser actions at each word in the form of incremental node count, which corresponds to the number of new nodes in a partial syntactic structure that are visited by the parser when incrementally integrating a word into the structure (complexity metric *C*; [16,30,31]). Depending on the parsing algorithm that is used, node count reflects the number of expand (top-down), reduce (bottom-up), or project (left-corner) actions between successive words (Table 1), all of which can be taken as roughly corresponding to the syntactic load or complexity of those words. Previous neuroimaging work has shown that node count effectively quantifies syntactic complexity in cognitive terms [16,19,21,22,27,28,32–35].

The left-corner strategy is thought to be cognitively plausible as a model of human language processing. It correctly predicts processing difficulty for center-embedded constructions [36–38], is compatible with a range of findings from the sentence processing literature [29], and accounts for brain activity during language processing [23,28]. However, much of the relevant psycholinguistic work has been done in English. A possible reason for the observation that left-corner parsing works well for English is that English phrases are strictly head-initial. The left corner of a phrase will therefore most often be its head, which determines the type of phrase (e.g., the head of a verb phrase (VP) is the verb) and which can be used to build structure predictively [39–41]. As English has grammatical properties that make it particularly well

**Table 1. Parser actions of top-down, bottom-up, and left-corner parsers for incremental parsing of the sentence "the boy sleeps."** The scan/shift action corresponds to processing or moving to the next word in the sentence and is not a structural operation. The other operations are explained in the text.

| Top-down | | Bottom-up | | Left-corner | |
|---|---|---|---|---|---|
| expand by | S ➜ NP VP | shift | the | shift | the |
| expand by | NP ➜ D N | reduce by | D ➜ the | project | D ➜ the |
| expand by | D ➜ the | shift | boy | project | NP ➜ D N |
| scan | the | reduce by | N ➜ boy | shift | boy |
| expand by | N ➜ boy | reduce by | NP ➜ D N | project | N ➜ boy |
| scan | boy | shift | sleeps | project | S ➜ NP VP |
| expand by | VP ➜ V | reduce by | V ➜ sleeps | shift | sleeps |
| expand by | V ➜ sleeps | reduce by | VP ➜ V | project | V ➜ sleeps |
| scan | sleeps | reduce by | S ➜ NP VP | project | VP ➜ V |

suited for left-corner parsing, these findings need not generalize to (the processing of) other languages. Indeed, it is quite likely that people's parsing strategies depend on the properties of the linguistic input, including the grammatical properties of the language in question. We therefore compare different parsing methods in terms of their ability to model syntactic processing of Dutch. In contrast to English, Dutch exhibits mixed headedness, with the verbal projections VP and IP being head-final. The left corner of a Dutch head-final VP will therefore often not be the verb, but rather a multi-word constituent, such as in sentence (1) below:

1. De student heeft een documentaire over het heelal gezien.
   the student has a documentary about the universe seen
   "The student has seen a documentary about the universe."

Notice the difference in word order between the Dutch example and its English translation. In English, the main verb precedes its object ("seen—a documentary about the universe"), thus yielding the head-initial order that is characteristic of English phrases. The reverse order is found in Dutch, where the verb follows the object (*een documentaire over het heelal–gezien*, "a documentary about the universe—seen"), giving the head-final order. The left-corner method predicts that the VP constituent in Dutch will be projected only after the entire preverbal NP object has been processed. This is unrealistically late, in particular if speakers of head-final languages adopt incremental or even predictive parsing strategies [42,43]. It might thus very well be that left-corner parsing is not the best strategy for Dutch structures. Dutch is typologically related to English, but the head-finality of its verb phrases might invite different processing strategies within the neural language network [44].

Note that our use of 3 different parsing methods does not imply that the human brain hosts multiple parsers. We use these methods as explicit and interpretable tools to derive values for incremental node count, a variable that counts the steps involved in syntactic structure building (e.g., the parser actions in Table 1). Thus, we use node counts as time-sensitive proxies for the operations that build structure in integratory or predictive ways, depending on how nodes (or, rather, parser actions) are counted. Neural responses associated with node count can therefore not be straightforwardly interpreted in terms of the representational properties of the syntactic structure being built. At present, we cannot distinguish between an interpretation of neural data in terms of evolving (syntactic) representations and an interpretation in terms of processing costs associated with incrementally deriving those representations. Thus, when node count derived from the top-down method significantly predicts brain activity, it does not mean that the resulting activity pattern reflects a partial, predictively activated syntactic representation, nor does it suggest that the brain is actually building hierarchical structure from the

top of the tree to the bottom. Rather, it suggests that Dutch sentence comprehension can be characterized via a predictive mechanism, which postulates structure early in time. If node counts derived from the bottom-up and left-corner methods additionally explain variance in brain activity, it means that hierarchical structure is also built in integratory and mildly predictive manners, respectively. Importantly, these strategies need not be mutually exclusive, and might even reflect one and the same mechanism. What makes them different is the completeness of the input they require to build structure, with the top-down method building structure maximally eagerly (using uncertain and incomplete input), while the bottom-up method requires complete input and is therefore minimally eager [24]. As different sentences and sentence positions vary in the extent to which they allow for predictive structure building, the comprehension process in full might thus be characterized in terms of multiple different processing strategies.

**1.1.2. From cognitive to brain states via temporal response functions.** The complexity metrics derived from different parsing models are regressed against electrophysiological brain activity through multivariate temporal response functions (mTRFs). TRFs are linear kernels that describe how the brain responds to a representation of a (linguistic) feature [45,46]. This approach is similar to that of recent neuro-computational fMRI studies, which use the canonical hemodynamic response function to fit syntactic predictors onto brain activity in a given region of interest [16,21,22,27,34]. But rather than assuming the shape of the response function, with the TRF method a response function can be estimated for each predictor separately, thus supporting time-resolved analyses. Moreover, by using multivariate TRFs, the acoustic properties of the auditory stimulus can be explicitly modeled, which is important for 2 main reasons. First, high-level linguistic features can be correlated with low-level stimulus properties, such that neural effects attributed to linguistic processing can also be explained as the brain's response to non-linguistic, acoustic information [47]. And second, because the neural response to acoustic properties is orders of magnitude larger than that to linguistic features [46,48,49], acoustic variance could mask subtle effects of linguistic information that are hiding in the data [33]. Such low-level factors have not always been modeled in previous work using neuro-computational models of syntactic structure building, limiting their interpretability (e.g., [18,19]). Conversely, TRF studies commonly analyze predictors that reflect lexical information at most; they are rarely extended to capture super-lexical features. To address these 2 limitations of previous work, the current study uses the TRF method to evaluate the brain's sensitivity to syntactic information in natural speech with high temporal precision, while appropriately controlling for lower-level factors.

## 1.2. The present study

In the current study, we compare different neuro-computational models in terms of their ability to predict source-reconstructed MEG activity of people listening to Dutch audiobook stories. The 3 models we evaluate rely on the same grammatical assumptions (X-bar theory), linking hypothesis (node count), and type of response function (TRF), but they differ in the parsing algorithm by which they build syntactic structure, and thus in the way they express syntax in time. Based on recent results linking syntactic processing to delta-band activity [32,33,35,50–54], and in line with the idea that the timescale of syntactic processing overlaps with the delta frequency range ([55,56]; see also Fig 3), we focus on MEG activity in this band.

Previous work has identified several brain areas in the left hemisphere that are responsive to complexity metrics derived from incremental parse steps, including the inferior frontal lobe [23,27,34], the anterior temporal lobe [16,21,23,27,28], and posterior superior and middle temporal areas [19,21–23,34]. Assuming that these effects reflect the operations involved in

syntactic structure building and that people employ similar parsing strategies when processing English and Dutch sentences, we expect effects in the same brain regions. Because the majority of this work has used fMRI, the timing of these effects is less clear, though the results of electrophysiological studies suggest that syntax in naturalistic language comprehension elicits distinct responses within the first 500 ms after word onset [15,28]. We therefore expect immediate effects of predictive and/or integratory structure building in this time window, but we adopt a larger analysis window to be able to capture potential responses related to subsequent revision or reanalysis [57]. In sum, the predictive accuracy of the different parsing models can give us insight into the processing strategies used by people comprehending Dutch (i.e., how they build structure), and the spatial-temporal properties of the effects provide clues about when and where in the brain these processes are implemented.

## 2. Methods

### 2.1. Participants

Twenty-four right-handed native speakers of Dutch (18 female, median age = 28, mean age = 33.4 years, age range = 20–58 years) were recruited via the SONA system of Radboud University Nijmegen. They all reported normal hearing, had normal or corrected-to-normal vision, and did not have a history of language-related impairments. Participants gave written informed consent to take part in the experiment, which was approved by the Ethics Committee for human research Arnhem/Nijmegen (project number CMO2014/288) and conducted in accordance with the Declaration of Helsinki. We note that the age range is rather wide, with 7 participants being older than 35, 4 of whom were above 50. Including these participants did not affect the results, which were qualitatively the same when we excluded them from our analyses.

### 2.2. Stimuli

The stimuli consisted of stories from 3 fairy tales in Dutch: 1 story by Hans Christian Andersen and 2 by the Brothers Grimm. All stories contain a rich variety of naturally occurring sentence structures, varying in syntactic complexity. In total, there are 8,551 words in 791 sentences, which are on average 10.8 words long (range = 1–35, SD = 5.95). They were auditorily presented in 9 separate segments, all of which were between 289 (4 min, 49 s) and 400 s long (6 min, 40 s; see S1 Table), for a total of 49 min and 17 s. The loudness of each audio segment was normalized using the Ffmpeg software (EBU R128 standard), and the resulting loudness level was the same for all participants. The transcripts of each story were automatically aligned with their corresponding audio recording using the WebMAUS segmentation software [58].

### 2.3. Syntactic annotations

We manually annotated syntactic structures for all sentences in the audiobooks following an adapted version of X-bar theory [25,26]. To be specific, we consistently created an X-bar structure for NPs and VPs, whereas intermediate projections for all other phrases were drawn only if they were needed to attach adjacent words to the structure (e.g., adjectival phrases (APs) were unbranched unless they were modified by an adverb or prepositional phrase). The X-bar template for NPs and VPs was strictly enforced in order to make a structural distinction between complements and adjuncts; complements were attached as sister of the head, while adjuncts were attached to an intermediate projection [25,26]. All phrases except for VPs and IPs are head-initial. An example of a hierarchical structure is given in Fig 2.

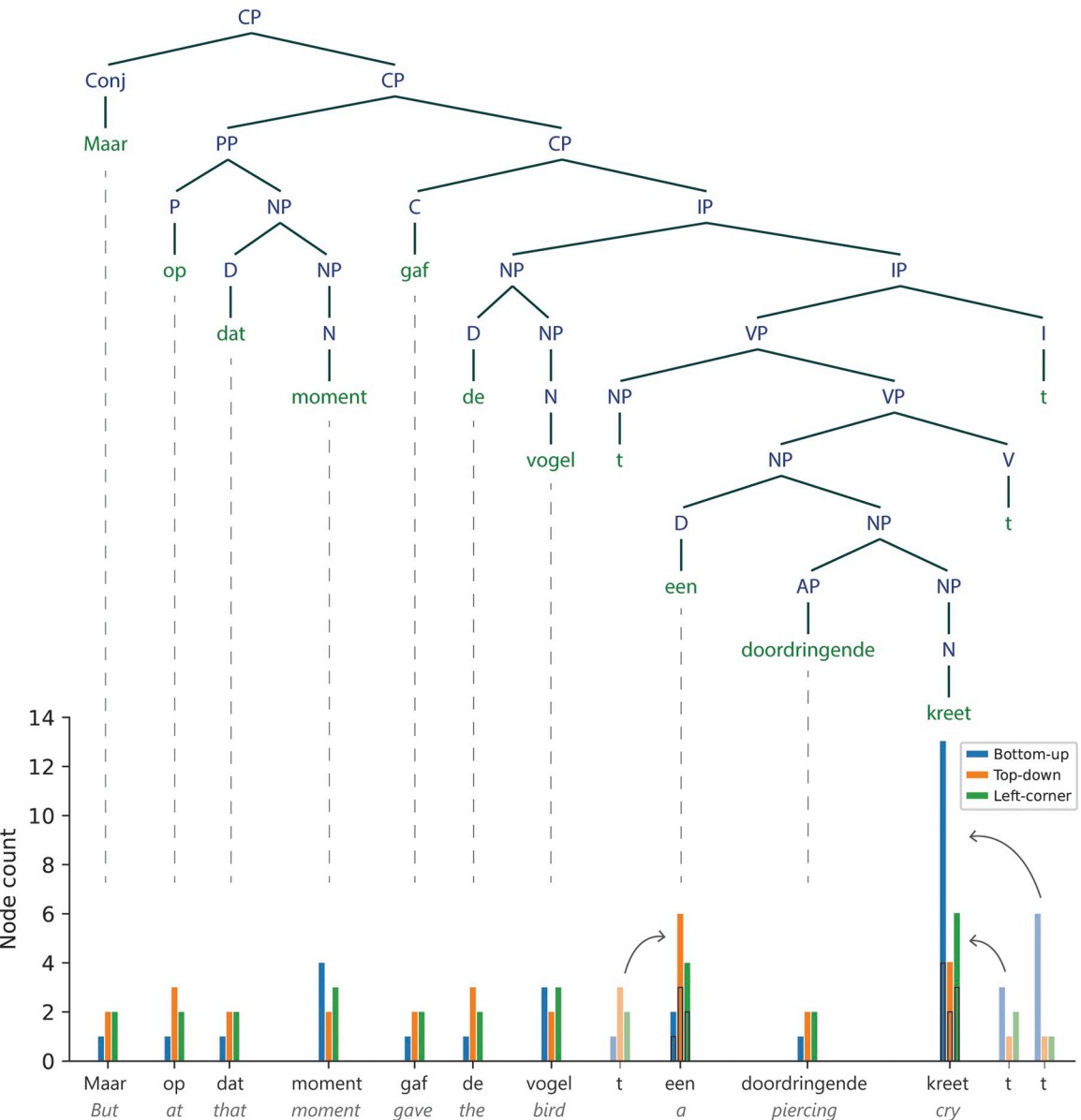

**Fig 2. Syntactic structure of an example sentence from one of the audiobook stories, with node counts for each terminal node presented below.** The *t* stands for trace and refers to the position at which the word or phrase to which it is related is interpreted. It does not have an acoustic correlate in the speech signal. As explained in the text, node counts for traces are added to node counts of the next word, except for traces that occupy sentence-final position, whose node counts are assigned to the previous word. Data are available on the Radboud Data Repository (https://doi.org/10.34973/m1vp-hc15).

Node counts were computed for each word in every sentence in 3 different ways. On the bottom-up strategy, a constituent node is posited when all daughter nodes have been encountered. This amounts to counting the number of closing brackets directly following a given word in a bracket notation. On the top-down strategy, a node is posited right before it is needed to attach the upcoming word to the structure. This amounts to counting the number of opening brackets directly preceding a given word in a bracket notation. And on the left-corner strategy, a node is posited when its left-most daughter node has been encountered. Terminal nodes were not included in the node count calculation.

As can be seen in Fig 2, the X-bar structures contain traces of movement, which are indicated by a *t* in the syntactic structure [25]. Movement is a syntactic operation adopted by certain syntactic theories to account for displacement (in psycholinguistics referred to as "filler-gap dependency"), the phenomenon whereby a word or phrase appears in a position different from its canonical position. Displacement is ubiquitous in the languages of the world and can be illustrated in English using a question like "what did you eat *t* yesterday?". Here, the word "what" is pronounced at the beginning of the sentence (position 1), but it is interpreted as the object of the verb "eat" (position 2, indicated by the *t* in the question). To account for the dependency between these 2 positions, some theories postulate empty elements, such as traces, at the canonical position. These are thought to be syntactically and semantically present even though they are not pronounced (hence the label "empty"). While the correct theoretical analysis of displacement is under debate, the existence of the phenomenon is not, and all (psycho) linguistic theories need to have a way to account for the observation that a single element can enter into multiple dependencies [59,60].

Because empty elements like traces do not have an acoustic correlate in the speech signal, we assigned their node counts to another word in the same sentence. Our reasoning was that the precise location of these elements could not be predicted with certainty, though it could be inferred after their putative position. We therefore added the node count of each empty element to the node count of the word following it. By doing so, we aimed to capture the syntactic processes associated with these elements (e.g., long-distance dependency resolution) around the times they occur. We will come back to this point in the discussion. The resulting node counts were time-aligned with the onsets of the words in the audio recordings. Each audio file could thus be represented as a vector with a node-count value at the onset of each word and zeros everywhere else. The vectors for each of the 3 parsing strategies were the predictors in the TRF analysis (see Section 2.7).

It has been suggested that the timescale of syntactic processing overlaps with the delta frequency range [55,56], which is consistent with the syntactic information rate in the 9 story segments that were used in this study. Syntactic information rates were derived from the time between the onsets of consecutive words with node counts above 2. Because we aim to illustrate the rate at which syntactic structure building can be initiated in the stimuli, only syntactically demanding words (i.e., node count >2) were included in the rate calculation. The probability density plots in Fig 3 show that the syntactic information rate for each parsing strategy in all story segments largely falls within the delta range. What this means is that the average time between 2 consecutive words that predict (top-down, left-corner) or integrate (bottom-up) multiple phrases is between 0.25 and 2 s. As such, the syntactic information rate provides information about the timescale at which syntax can be perceptually inferred from naturalistic speech. To be clear, it should not be understood as reflecting the average lengths of phrases or the number of phrases per unit of time. Syntax builds hierarchical structures, so sentence structures cannot be described in terms of these sequential properties. Rather than being concatenated, phrases are hierarchically embedded in one another. The sequential length of phrases is therefore extremely variable, and there is no regular phrase rate in natural language [61,62]. We therefore think that it is not accurate to define the rate of syntactic complexity as the number of phrases per time unit. Here, we instead derive the syntactic information rate from the time between syntactically demanding words, as encoded in the different node-count vectors. It therefore reflects the general rate at which predictive or integratory structure building can be initiated. And because it bears no relation to the sequential lengths of phrases, our claim that the syntactic information rate falls in the delta range does not mean that each phrase can be covered by a delta cycle.

**Fig 3. Probability density functions showing that the average rate at which syntactically relevant information is presented largely falls within the 0.5–4 Hz delta band (highlighted in grey) in all story segments.** The dashed vertical lines reflect the median rate per parsing method. Data are available on the Radboud Data Repository (https://doi.org/10.34973/m1vp-hc15).

## 2.4. Procedure and data acquisition

Participants were individually tested in a magnetically shielded room. They were instructed to attentively listen to the 9 audiobook stories while sitting still and looking at a fixation cross that was presented in the middle of the screen. After each of the 9 story blocks, 5 multiple-choice comprehension questions (each with 4 options) were asked. On average, participants answered 88.1% of the questions correctly (SD = 7.52%), showing that they paid attention to and understood the content of the stories.

The MEG data were recorded with a 275-channel axial gradiometer CTF system at a sampling rate of 1,200 Hz. The audio recordings were presented using Psychtoolbox in MATLAB [63] via earphones inserted into the ear canal. Participants' eye movements and heartbeat were recorded with EOG and ECG electrodes, respectively. Throughout the recording session, their head position was monitored using 3 head localization coils, one placed in fitted earmolds in each ear and one placed at the nasion [64]. Each block started with a 10-s period during which resting state data were recorded. In the break between story blocks, participants were instructed to reposition their head location in order to correct for head movements. After the MEG session, each participant's head shape was digitized using a Polhemus 3D tracking device, and their T1-weighted anatomical MRI was acquired using a 3T Skyra system (Siemens).

## 2.5. MEG preprocessing

Preprocessing was done using MNE-Python (version 0.23.1). The MEG data were first down-sampled to 600 Hz and band-pass filtered at 0.5–40 Hz using a zero-phase FIR filter (MNE-Python default settings). We then interpolated channels that were considered bad using Maxwell filtering and used Independent Component Analysis to filter artifacts resulting from eye movements (EOG) and heartbeats (ECG). We segmented the data into 9 large epochs, whose onsets

and offsets corresponded to those of the audio recordings. Source reconstruction was done for each epoch separately.

## 2.6. Source reconstruction

Individual head models were created for each participant with their structural MRI images using FreeSurfer [65]. The MRI data were then co-registered to the MEG with MNE co-registration, using the head localization coils and the digitized head shape. We set up a bilateral surface-based source space for each individual participant using 4-fold icosahedral subdivision, resulting in 2,562 continuous source estimates in each hemisphere. The forward solution was computed using a BEM model with single layer conductivity. We low-pass filtered the signal at 4 Hz using a zero-phase FIR filter (corresponding to the 0.5–4 Hz delta band) and estimated sources using the dSPM method (noise-normalized minimum norm estimate), with source dipoles oriented perpendicularly to the cortical surface. The noise covariance matrix was calculated based on the resting state data that were recorded before each story (all concatenated). Before TRF analysis, each source estimate was downsampled to 100 Hz to speed up further computations.

## 2.7. Predictor variables

To control for brain responses to acoustic information, all models included 2 acoustic predictors: an eight-band gammatone spectrogram (i.e., envelope of the acoustic signal in different frequency bands) and an eight-band acoustic onset spectrogram. Both spectrograms covered frequencies from 20 to 5,000 Hz in equivalent rectangular bandwidth space [66] and were resampled to 100 Hz to match the sampling rate of the MEG data. The onset spectrogram was derived from the gammatone spectrogram using an auditory edge detection model [67] implemented in Eelbrain (Version 0.37.3; [68]).

All models also included 4 word-based predictors that are all strongly linked to brain activity during naturalistic language processing [45,46,69]. These were word rate and the 3 statistical predictors word frequency, surprisal, and entropy. All predictors were modeled on the gammatone predictors in terms of length and sampling rate.

The word rate predictor is simply a one-dimensional array with the value 1 at word onsets and the value 0 everywhere else.

The frequency of word $w$ was computed by the taking negative logarithm of the number of occurrences of $w$ per million words (varying between 0 and 1), extracted from the SUBTLEX-NL database of Dutch word frequencies [70]:

$$Word\ frequency(w) = -log_2(frequency(w)).$$

Word frequency was represented via the negative logarithm, because in this way infrequent words will get high values and frequent words will get low values, in line with the brain response to word frequency (i.e., a larger response to infrequent words; [21,45]). For some words, we could not compute a frequency value because the word did not appear in the database. Manually checking them revealed that these were uncommon (and thus likely infrequent) words, so we assigned to them the value corresponding to the lowest frequency of all words present in the audiobook.

Surprisal is the conditional probability of a word given the preceding linguistic context, quantified as the negative log of this probability [29,71]. Thus, the surprisal of word $w$ at position $t$ is calculated via:

$$I(w_t) = -log_2(P(w_t|context)).$$

**Table 2. Predictors included in each model.**

| Predictor(s) / Model name | Spectrogram/onsets | Word onset | Word frequency/entropy/surprisal | Bottom-up | Top-down | Left-corner |
|---|---|---|---|---|---|---|
| Bottom-up + Top-down | X | X | X | X | X | |
| Bottom-up + Left-corner | X | X | X | X | | X |
| Top-down + Left-corner | X | X | X | | X | X |
| Full | X | X | X | X | X | X |

Word surprisal was computed from conditional probabilities obtained with GPT-2 for Dutch [72]. GPT-2 calculates surprisal for sub-word tokens, which in the majority of cases correspond to single words. However, long words are tokenized into multiple sub-word tokens. In those cases (490 words, 5.73% of the entire stimulus set), we used the estimated surprisal of the first token as an index of the surprisal of the entire word. GPT-2 used the preceding 30 words (or, rather, sub-word tokens) as context, so context in the formula above refers to $(w_{t-30} \ldots w_{t-1})$.

Entropy at word position $t$ is the uncertainty before observing the next word $w_{t+1}$ given the preceding context. Context was again defined as the previous 30 words (including $w_t$), and conditional probabilities were again obtained with GPT-2 for Dutch [72]. Entropy at word position $t$ was then calculated as the sum of the conditional probabilities of each next word (within the set of possible upcoming words $W$), weighted by the negative logarithm of this probability:

$$H(t) = -\sum_{w_{t+1} \in W} P(w_{t+1}|context) log_2(P(w_{t+1}|context)).$$

Our syntactic models included the syntactic predictors bottom-up node count, top-down node count, and left-corner node count. We constructed a total of 4 models (see Table 2).

### 2.8. Model estimation

TRFs were estimated for each subject and MEG source point separately using Eelbrain (Version 0.37.3; [68]). The MEG response at time $t$, denoted as $\hat{y}(t)$, was predicted jointly by convolving each TRF with a predictor time series shifted by K time delays [46,68]:

$$\hat{y}(t) = \sum_{f=1}^{F} \sum_{k=1}^{K} \beta_f(\tau_k) x_f(t - \tau_k).$$

Here, $x_f$ is the predictor time series and $\beta_f(\tau_k)$ is the coefficient of the TRF of the corresponding predictor at delay $\tau_k$. The coefficient of the TRF at delay $\tau$ thus indicates how a change in the predictor affects the predicted MEG response $\tau$ milliseconds later. To generate each TRF, we used 50-ms wide Hamming windows and shifted the predictor time series between −100 and 1,000 ms at a sampling rate of 100 Hz, thus yielding K = 110 different delays. The length of the TRF was chosen based on the latency of syntactic effects in naturalistic paradigms [15,18] and classical event-related potential (ERP) violation paradigms [57]. Before estimating the TRF, all predictors as well as the MEG data were mean-centered and then normalized by dividing by the mean absolute value.

TRFs were estimated using a 5-fold cross-validation procedure. We first concatenated the data for each subject along the time axis and then split them up into 5 equally long segments. During each cross-validation run, 3 segments were used for training, one for validation, and one for testing. For each test segment, there were 4 training runs with each of the remaining segments serving as the validation segment once. Using a boosting algorithm [73] to minimize

the l1 error, one TRF was estimated for each of the 4 training runs (selective stopping based on l1 error increase). The resulting 4 TRFs were averaged to predict responses in the test segment. This analysis yields an average TRF for each predictor in each model, as well as a measure of reconstruction accuracy for the whole model. Reconstruction (or predictive) accuracy refers to the fit between the predicted and the observed MEG signal at each source point, quantified in terms of explained variance in $R^2$. Reconstruction accuracy can be seen as a measure of neural tracking: the larger the reconstruction accuracy for a given model, the more closely the brain tracks the predictors in that model.

## 2.9. Model comparison

We first tested whether separately adding each of the 3 syntactic predictors to a base model with all control predictors (see S2 Table) would increase the reconstruction accuracy. Having established this (see S1 Text, section 1), we then determined the unique contribution of each syntactic predictor by comparing the reconstruction accuracy of the full model to the reconstruction accuracy of a null model from which only one of the predictors was omitted. The 3 null models we evaluated were Bottom-up + Top-down, Bottom-up + Left-corner, and Top-down + Left-corner (see Table 2). Comparing their reconstruction accuracy to the accuracy of the full model yields an accuracy difference measure for the left-corner, top-down, and bottom-up predictors, respectively. This comparison thus tests whether a predictor explains variance in the brain signal above and beyond the variance explained by all other predictors.

To determine where in the brain the reconstruction accuracy of the full model was different from that of the null model, we smoothed the source points of both models separately (Gaussian window, SD = 14 mm) and tested for differences in their source maps using nonparametric cluster-based permutation tests [74]. For all contrasts between full and null model, we applied two-tailed paired-samples *t* tests at each source point, clustered adjacent source points (minsource = 10) with an uncorrected *p*-value lower than 0.05, and evaluated clusters of activity by comparing their cluster-level test statistic (sum of individual t-values) to a permutation distribution. The permutation distribution was generated based on the maximum cluster-level t-value in each of 10,000 random permutations of the same data, in which the condition labels were shuffled within subjects. The significance of clusters was evaluated at an alpha value that was Bonferroni-corrected for the number of tests (alpha = $0.05/n_{tests}$). As an estimation of the effect size of the significant clusters, we report $t_{av}$, which corresponds to the average t-value within the significant cluster (the cluster-level t-value divided by the number of significant source points). When multiple clusters are significant, we report the test statistic of the cluster with the largest number of significant source points.

The syntactic predictors are positively correlated. This is mainly the case for the left-corner predictor, whose Pearson correlation with the bottom-up and top-down predictors is 0.74 and 0.80, respectively (S1 Fig). A high correlation between predictors in a regression analysis can lead to multicollinearity, which can in turn result in increased variance of their TRF coefficients [69]. This issue typically emerges when the variance inflation factor (VIF) is above 5, which indicates that the variance in one predictor can be explained by a linear combination of the other predictors [75]. We computed the VIF for each predictor in the full model by taking the diagonal of the inverse of the correlation matrix shown in S1 Fig. This showed that when all predictors are included, the VIF for both top-down ($VIF_{top-down}$ = 5.09) and left-corner ($VIF_{left-corner}$ = 9.75) is above 5, indicating that multicollinearity between the predictors might hinder TRF coefficient estimation.

As a control, we therefore repeated our analyses with TRF models in which the VIF is below 5 for all predictors. In this control analysis, we evaluated whether the addition of each of

the 3 syntactic predictors to a base model, which included no syntactic predictors at all (S2 Table), improves reconstruction accuracy. This analysis is conceptually similar to the analysis reported in the main manuscript, but it does not take into account co-dependencies between the different syntactic predictors because these predictors never appear in the same model. The results of this analysis are both qualitatively and quantitively very similar to the results of the main analyses (see S2–S4 Figs and S1 Text, section 1), supporting the conclusion that the different predictors explain unique variance in the MEG data despite being positively correlated.

## 2.10. Evaluation of the response functions

In addition to analyzing the fit between the predicted and the observed signal, we also evaluated the estimated response function, which provides information about the temporal relationship between the predictor and the neural response. This analysis involves the absolute value of the coefficients of the TRF at each time and source point (sources smoothed by a Gaussian window, SD = 14 mm), baseline-corrected by subtracting the average activity at negative lags (i.e., 100 ms preceding stimulus onset at t = 0). If the coefficients for a given predictor are significantly above zero, this indicates that the brain responds to the information encoded in that predictor. In a spatiotemporal cluster-based permutation analysis, we first applied one-tailed one-sample $t$ tests at each source-time point to determine whether the rectified TRF coefficients deviate from zero. The t-values of adjacent source-time points (minsource = 40, mintime = 40 ms) with an uncorrected $p$-value lower than 0.025 were then summed, and their cluster-level test statistic was compared to a permutation distribution based on 10,000 random permutations of the same data. The significance of clusters was evaluated at an alpha value that was Bonferroni-corrected for the number of tests.

## 2.11. Prosodic control analysis

Neural activity in the delta band tracks prosodic modulations in speech [76–78]. And because syntactic structure is related to prosodic structure, and syntactic and prosodic variation are temporally correlated [79–81], recent studies have argued that delta-band responses to syntactic structure might be explained in terms of prosody (for a discussion, see [82]). To show that neural activity is uniquely attributable to syntactic processing, it is thus important to explicitly control for prosodic properties of the speech signal. We therefore conducted a prosodic control analysis in which we included prosodic boundary strength as regressor in our TRF models.

The prosodic boundary at each word was computed using the Wavelet Prosody Toolkit [83]. This algorithm estimates prosodic boundaries using a continuous wavelet transform (CWT) of a combined signal that is composed of fundamental frequency, energy, and word duration. We first estimated fundamental frequency, energy, and duration, combined the time courses of these 3 features, and performed a CWT of the combined signal. From the output of the CWT (a decomposition into different scales), the algorithm finds the minima across the scales, which provide a continuous measure of prosodic boundary strength for each individual word.

In our stimuli, the strength of prosodic boundaries (PBs) is positively correlated with the strength of syntactic boundaries, quantified by node count. But the relationship between syntax and prosody is dependent on how node count is computed: PB is positively correlated with bottom-up node count (Pearson r = 0.34) and left-corner node count (r = 0.19), but not correlated with top-down node count (r = 0.04). The positive correlation between PB strength and bottom-up node count is in line with the (psycho)linguistic literature, which has shown a

relation between the strength of prosodic boundaries and the strength of closing syntactic boundaries [79–81]. In order to evaluate whether any of the syntactic effects we found could be explained in terms of brain responses to prosody, we included prosodic boundary strength as predictor in our TRF models. The results are reported in S1 Text, section 3 and shown in S6 Fig.

## 3. Results

### 3.1. Model comparison

Using cluster-based permutation tests in source space, we tested where in the brain the reconstruction accuracies were modulated by each of the 3 syntactic predictors. Clusters of activity were evaluated at alpha = 0.0083 ($n_{tests}$ = 3 accuracy differences * 2 hemispheres). All predictors significantly improve reconstruction accuracy, with clusters mostly in the left hemisphere (see Fig 4A–4C). The improvement is largest for the top-down predictor, which explains variance in many regions of the left hemisphere ($t_{av}$ = 5.49, $p$ = 0.0034), as well as in an anterior

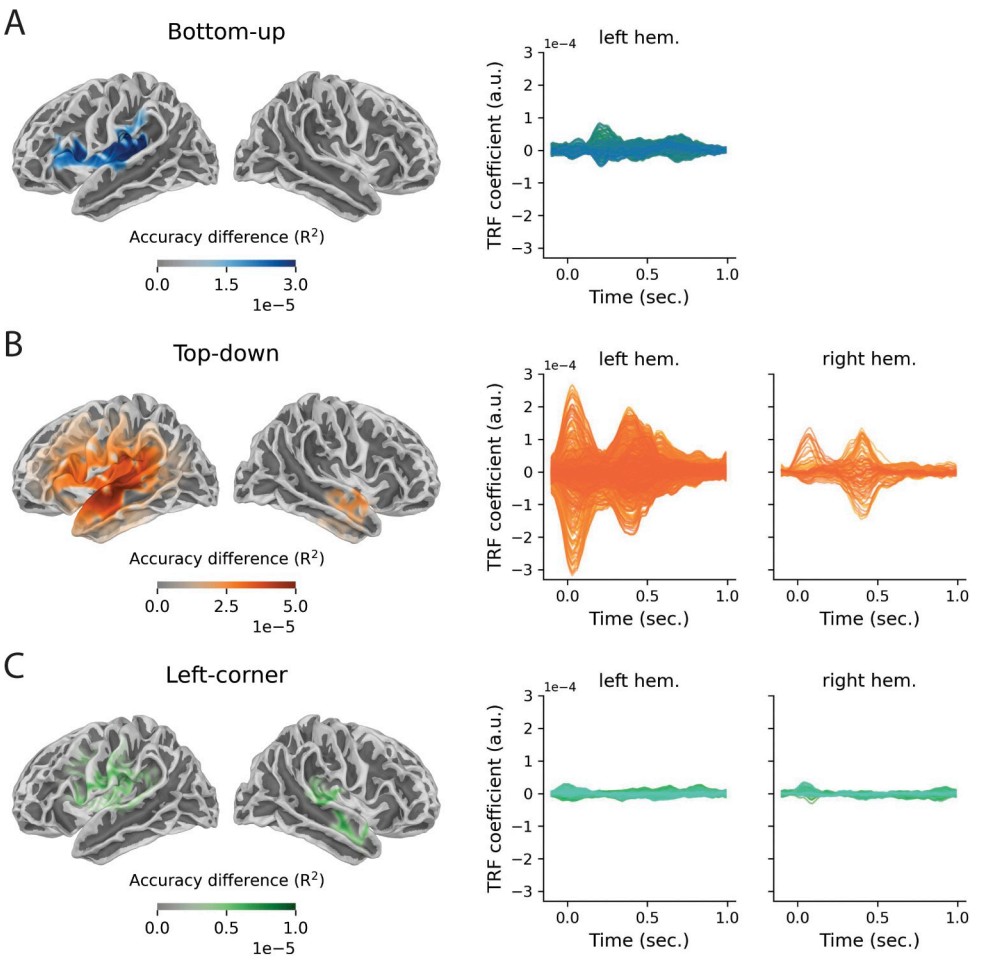

**Fig 4. Sources of significantly improved explained variance and temporal response functions in significant source points.** Results are shown separately for the effects of the bottom-up (**A**), top-down (**B**), and left-corner (**C**) predictors. Significance was determined by comparing the reconstruction accuracy of the full model to the reconstruction accuracy of a null model from which the relevant predictor was omitted. All clusters that were significant at uncorrected alpha = 0.05 are displayed. Notice that the scales of the color bars are different across the source plots. Data are available on the Radboud Data Repository (https://doi.org/10.34973/m1vp-hc15).

part of the right temporal lobe ($t_{av}$ = 2.60, $p$ = 0.0075). The strongest left-hemispheric effects are found in superior and middle temporal regions and in inferior and middle frontal regions. The bottom-up predictor similarly engages inferior frontal and temporal regions, but only one cluster around Heschl's gyrus is significant at the adjusted alpha level ($t_{av}$ = 3.84, $p$ = 0.0081). Last, the left-corner predictor improves reconstruction accuracy in an area at the border of the temporal and frontal lobe in both the left ($t_{av}$ = 5.72, $p$ = 0.0057) and the right hemisphere ($t_{av}$ = 3.35, $p$ = 0.0081). It is noteworthy that even though the 3 syntactic predictors are positively correlated with one another, each of them explains unique variability in the MEG data that could not be attributed to the other 2 syntactic predictors.

### 3.2. Evaluation of the response functions

Given that all of the predictors increase reconstruction accuracy, we examined their response functions, which reveal a more detailed picture of the neural time course of syntactic structure building. Fig 4 shows the TRFs within the significant regions of the cluster-based source analysis of reconstruction accuracies, in the left and the right hemisphere separately. These plots yield an estimate of the magnitude of the brain response to the syntactic predictor at each time point. All TRFs come from the full model, in which all predictors are competing for explaining variance, so the increases in amplitude reflect components of the neural response that are best explained by the respective predictor. In line with the reconstruction accuracy results, the TRF amplitudes clearly reveal that the neural response to the information encoded in top-down node counts is stronger than the response to node counts derived from the bottom-up or the left-corner method. The top-down TRF peaks twice within the first 500 ms, while the bottom-up TRF has a more temporally spread time course.

While these results are informative about the overall strength and timing of the responses, they do not show which regions are involved at which time points. This information is provided in Fig 5, which shows the source t-values (based on one-tailed, one-sample $t$ tests) of the rectified TRFs of the 3 syntactic predictors. Nonparametric cluster-based permutation tests were used to determine when and where the TRF coefficients of each syntactic predictor deviated from zero. Because this involves 6 comparisons (3 TRFs * 2 hemispheres), clusters were evaluated at alpha = 0.0083. The results are split up into 5 time windows (corresponding to different delays in TRF estimation) for the purpose of visualization only. They show that the effect

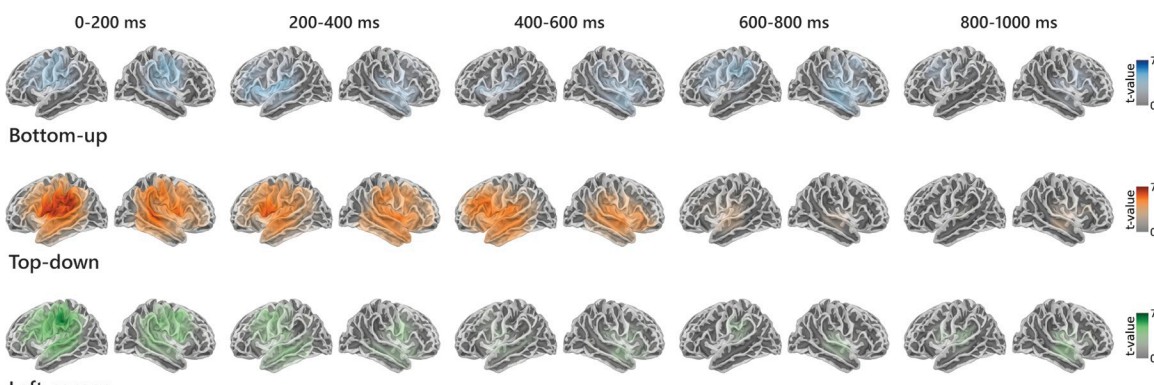

**Fig 5.** Sources of the TRFs for node count derived from bottom-up, top-down, and left-corner parsers, representing early to late responses. The colors represent t-values in source clusters that were significantly responsive (at corrected alpha = 0.0083) to the predictor in the indicated time windows. The midpoints of the color scales are at 70% of the maximum, such that the colored plots highlight the sources with the strongest effects. Data are available on the Radboud Data Repository (https://doi.org/10.34973/m1vp-hc15).

of the top-down predictor is strongest and peaks early (as in Fig 4B), but also that it is distributed rather bilaterally at those early lags. That suggests that predictive structure building, as quantified by top-down node counts, poses strong demands on brain computation, in particular in early time windows. The fact that right-hemisphere activity contributed less to improving reconstruction accuracy might however indicate that these effects are less stable across participants. Another potentially informative observation in Fig 5 is that the left-corner TRF seems to combine aspects of both the top-down and the bottom-up effects. It peaks early, like the top-down TRF, but its time course is prolonged, which is the case for the bottom-up TRF as well. This observation is interesting because it aligns with the idea that the mildly predictive left-corner parsing strategy combines properties of top-down and bottom-up parsing.

### 3.3. Region of interest analysis

To further explore the spatiotemporal differences between the response functions of the syntactic predictors, we analyzed the TRFs in 3 specific regions of interest (ROIs) that have been linked to syntactic structure building in naturalistic contexts: the inferior frontal gyrus (IFG), posterior temporal lobe (PTL), and anterior temporal lobe (ATL) in the left hemisphere (Fig 6A). All of these ROIs also showed up in one or more of the contrasts in the accuracy analysis (Fig 4) and the TRF analysis (Fig 5). The locations of these ROIs were defined based on the peak coordinates in Montreal Neurological Institute (MNI) space of an fMRI study on syntactic structure building [84]. We used these coordinates as seeds to create spheres with a 40-mm radius for the temporal pole (MNI coordinates: −48, 15, −27) and posterior superior temporal sulcus (−51, −39, 3). In addition, 30-mm spheres were created around the coordinates of the left inferior frontal cortex, which included both the pars triangularis (−51, 21, 21) and the pars orbitalis (−45, 33, −6). To compute the reconstruction accuracies and TRFs per ROI, we averaged the reconstruction accuracies and rectified TRFs over each of the region's source points.

The results in Fig 6B show for each ROI the improvement in reconstruction accuracy when the syntactic predictors are separately added to the null model. Effects of the top-down predictor are strongest in general, and in particular in the PTL, where all subjects show evidence of responses associated with top-down node counts. Supporting this impression, two-tailed paired-samples $t$ tests (with alpha = 0.0056, Bonferroni-corrected for 9 tests) reveal that adding the top-down predictor to the null model improves the reconstruction accuracy in all 3 ROIs (IFG: t(23) = 5.54, $p < 0.001$; PTL: t(23) = 5.04, $p < 0.001$; ATL: t(23) = 3.50, $p = 0.0020$). The addition of the bottom-up predictor seems to improve reconstruction accuracy in the IFG, but this effect did not survive multiple comparisons correction, t(23) = 2.73; $p = 0.012$. Cluster-based permutation tests (with alpha = 0.0056) were then used to compare the time courses of the syntactic predictors in each ROI. As shown in Fig 6C, the TRF of the top-down predictor has a higher amplitude than the TRFs of both the bottom-up and left-corner predictors in each ROI. In addition, the bottom-up response is stronger than the left-corner response in the IFG, corroborating the reconstruction accuracy results. What is also noteworthy is that the top-down TRF is bimodal in all ROIs, peaking at around 100 and 400 ms, and that the strongest peak of the bottom-up TRF falls in between those 2 peaks, around 300 ms, in the IFG and ATL.

### 4. Discussion

In this study, we investigated when the brain projects its knowledge of syntax onto speech during natural story listening. Atemporal syntactic structures were extended into the time domain via the use of incremental node count, whose temporal distribution showed that the delta band

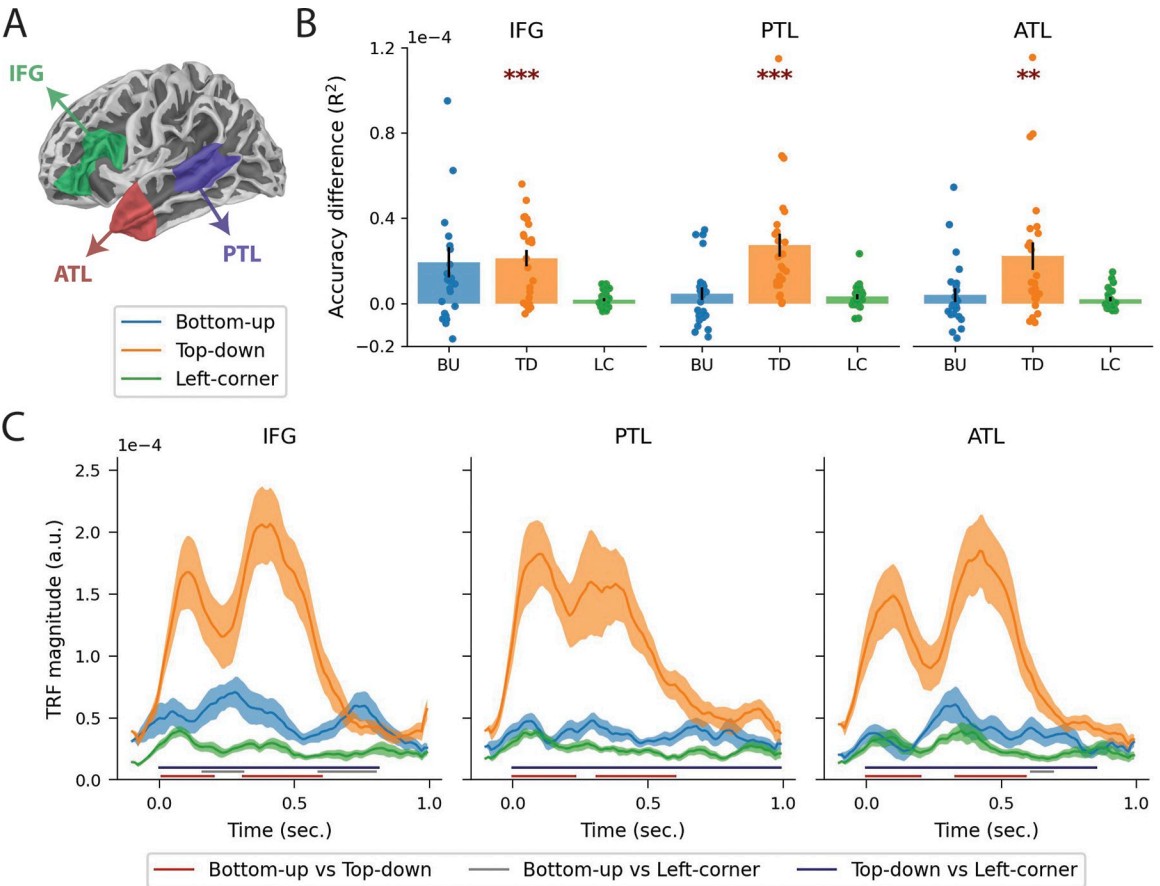

**Fig 6. ROI analysis.** (A) Spatial extensions of the 3 regions of interest. (B) Difference in reconstruction accuracy with the full model, plotted for left IFG, PTL, and ATL. The labels on the x-axis refer to the syntactic predictors that were taken out of the full model, so the height of each bar indicates the reduction in reconstruction accuracy compared to the full model when only that predictor is omitted. The drops represent the accuracy difference for individual participants, and the error bars represent the standard error of the mean across subjects. (C) Temporal response functions for node count derived from bottom-up, top-down, and left-corner parsers in the full model. Error bars reflect the standard error of the mean per time sample. The horizontal bars below the TRFs reflect the temporal extensions of the significant clusters from the pairwise analysis. Data are available on the Radboud Data Repository (https://doi.org/10.34973/m1vp-hc15). ATL, anterior temporal lobe; IFG, inferior frontal gyrus; PTL, posterior temporal lobe; ROI, region of interest.

is a syntactically relevant timescale in our stimuli. Using a forward modeling approach to map node counts onto source-reconstructed delta-band activity, we then compared three parsing models that differ in the dynamics of structure building. A key finding of these analyses is that neural source dynamics most strongly reflect node counts derived from a top-down parsing model. This model postulates syntactic structure early in time, suggesting that predictive structure building is an important component of Dutch sentence comprehension. The additional (though weaker) effects of the bottom-up and left-corner predictors indicate that integratory and mildly predictive parsing also play a role, and suggest that people's processing strategy might be flexibly adapted to the specific properties of the linguistic input.

## 4.1. Predictive structure building in the brain

Node counts derived from all 3 parsing models explained unique variance in delta-band MEG activity, consistent with recent studies showing a relationship between delta-band activity and syntactic processing [32,33,35,50–55]. The results also align with prior studies that use neuro-

computational language models to study syntactic processing [16,19,21–23,27,28,34], but they nevertheless advance our understanding of the neural encoding of syntax due to the inclusion of low-level (linguistic and non-linguistic) predictors in our TRF models. In addition to a number of acoustic predictors, the null models also contained information-theoretic predictors (e.g., surprisal, entropy) that partially reflect syntactic information and therefore explain some of the variance in brain activity whose origin is syntactic. By including these semi-syntactic control predictors, we stacked the cards against us and likely underestimated the "true" neural response to syntactic structure. To determine to what extent our syntactic effects are influenced by the presence of these control predictors, we ran an additional analysis with TRF models from which surprisal, entropy, and word frequency were omitted. Removing these statistical predictors increases the variance explained by each syntactic predictor, but the pattern of results remains unchanged: the top-down model still outperforms the 2 other parsing models (see S5 Fig and S1 Text, section 2). The fact that the effects of syntactic processing are robust and stable, both in the presence and in the absence of statistical control predictors, underscores the relevance of syntax for the neural mechanisms underlying language comprehension [35].

Of all syntactic predictors, node counts derived from a top-down parsing method were the strongest syntactic predictor of brain activity in language-relevant areas. These effects peaked twice within the first 500 ms after word onset and encompassed mostly superior and middle temporal, and inferior and middle frontal areas in the left hemisphere. The predictiveness of top-down node counts is somewhat at odds with previous studies that have looked at different parsing models in naturalistic comprehension, which either find that top-down methods are less predictive of brain activity [23,34,85] or that they do not differ from other parsing methods [21]. What could account for the strong top-down effects? One explanation is that top-down node counts capture the predictive nature of language processing well. There is substantial evidence from psycholinguistics that people generate structural predictions across a variety of syntactic constructions [39,86–89], and they do so in naturalistic contexts as well [15,18,90]. These predictive structure-building processes are mostly associated with activity in the left PTL [21,23,91–93] and left IFG [91], both of which were responsive to the top-down predictor in the current study. Matchin and colleagues [92] suggest that the PTL is involved in predictive activation of sentence-level syntactic representations and/or increased maintenance of the syntactic representations associated with lexical items when they are presented in a sentential context (see also [93]). On both interpretations, the PTL encodes structural representations that can be activated in a predictive fashion and are later to be integrated with the sentence-level syntactic representation in IFG [94–96]. Importantly, this process does not proceed in a purely feedforward manner, but rather relies on recurrent connections between temporal and frontal regions [97]. That the TRF of the top-down predictor is bimodal is consistent with this idea (Figs 4B and 6C). The first peak around 100 ms was strongest in posterior regions and likely reflects predictive structure building, which can occur in the PTL within the first 150 ms after word onset [92]. Indeed, the early timing of this effect is consistent with the predictive nature and temporal spell-out of top-down parsing, which is maximally eager and postulates syntax early in time. Previous electrophysiological studies have shown that phrase-structure violations elicit similarly early responses, including an early left anterior negativity [98–100] and a syntactic Mismatch Negativity [101,102], which have been linked to left superior temporal cortex in lesion and localization studies [98–100,102]. The fact that phrase-structure violations are detected so early strongly suggests that the brain generates predictions about the syntactic properties of upcoming material, perhaps in the form of precompiled phrase structure representations [94,98,101]. The second, later peak in the top-down TRF was strongest in more anterior regions and is largely consistent with syntactic surprisal effects, both in terms of timing [15,18,90] and spatial distribution [103]. It might therefore reflect responses related to the

disconfirmation of predicted structures based on incoming information, which triggers a revision of the initial parse of the sentence. An additional observation is that the effects of the top-down predictor, and to a lesser extent also those of the left-corner predictor, are somewhat bilateral. A tentative explanation for this finding is that syntactic prediction, as quantified by top-down and left-corner node counts, is demanding and therefore requires support from right-hemispheric regions. These areas are presumably not the locus of the syntactic representations and computations themselves but might be activated when processing demands are increased [104]. Understanding how these MEG time courses are related to the structural neuroanatomy of syntax is an important topic for future research, which would benefit from a multi-methodological approach.

Compared to the top-down TRF, bottom-up effects were relatively weak in amplitude. This is quite striking because, of all syntactic features, the bottom-up predictor is most strongly correlated with properties of the acoustic signal. That is, bottom-up node count quantifies syntactic phrase closure, which is related to the closure of prosodic phrases [79–81] and moderately correlated with the strength of prosodic boundaries across all story segments (r = 0.34 in our stimuli). To determine whether any of the syntactic effects might be explained in terms of brain responses to prosody, which also modulates delta-band activity [76–78], we included prosodic boundary strength as predictor in our TRF models. The results show that while the top-down effects are quite stable, the unique variance explained by the bottom-up and left-corner predictors is significantly reduced by the addition of prosodic boundary strength as regressor (see S6 Fig and S1 Text, section 3). It thus seems that the integratory effects of bottom-up and left-corner node count are modulated by prosody, pointing to a potential role for prosody in supporting the inference of syntactic structure [82,105]. The top-down effect, however, is not affected by prosodic boundary strength, providing additional support that it reflects abstract structure building. More generally, this result reinforces the idea, alluded to the introduction, that perception requires the brain to add information to its analysis of the physical environment. Spoken language comprehension is more than passive extraction of information from speech—it is an active inference process, during which the brain adds its knowledge of syntactic structure. Because syntactic effects cannot be driven entirely by what is in the signal, they must at least partially reflect what the brain perceptually infers from that signal.

It seems that the bottom-up and top-down effects are inversely related. Strong top-down effects, which indicate predictive processing, are accompanied by weak bottom-up effects, and vice versa. One of the advantages of prediction in language comprehension is that it can reduce the burden on future integration processes [106]. If a structural representation has already been pre-built or pre-activated, correctly predicted incoming words only have to be inserted into the existing structure, so integration costs for these words are low. Because integration costs are approximated via bottom-up node count, any effects of bottom-up node count should be reduced if people successfully engage in predictive processing. This account would thus predict that when top-down metrics modulate brain activity for a given sentence, bottom-up metrics will not provide a good fit for that same sentence, and conversely, when top-down metrics do not provide a good fit, bottom-up metrics should be highly predictive. While it should be investigated in future work whether the effects of top-down and bottom-up metrics indeed go hand in hand in this anti-correlated way, the results of 2 naturalistic studies with spontaneous speech are consistent with this possibility. Contrasting with coherent audiobook narratives, spontaneously produced speech contains dysfluencies and corrections, which might make participants less inclined to rely on predictive processing. In an fMRI study by Giglio and colleagues [34], English speakers had to listen to other people's verbal summaries of a TV episode. The authors found that bottom-up node counts modulated activity in language-relevant brain areas (LIFG and LPTL) more strongly than top-down node counts. Second, an EEG

study by Agmon and colleagues [107] investigated the neural encoding of different linguistic features when Hebrew speakers listened to a spontaneously generated narrative. Using a TRF analysis on low-frequency activity, they found that the neural response for words closing a clause (bottom-up) was stronger than the neural response for words opening a clause (top-down). Both studies indicate that in the comprehension of spontaneous speech the brain relies relatively strongly on integratory processing. A possible implication of these findings is that parsing strategies can be flexibly adapted to the specific properties of the current linguistic input (e.g., grammatical properties, sentence complexity, reliability of predictive cues), such that people are less likely to engage in predictive structure building if the input contains ungrammatical sentences that make predicting ineffective [106,108]. As an exploratory analysis of the potential trade-off between predictive and integratory structure building, we investigated whether predictability (quantified through surprisal) modulates the demands on bottom-up structure building (see S7 Fig and S1 Text, section 4; see also [85]). How exactly the demands on predictive and integratory structure building dynamically change over the course of sentence processing should be investigated in future research.

Effects of the left-corner predictor were also relatively weak compared to effects of the top-down predictor, especially after the addition of prosodic boundary strength as control regressor (S6 Fig). The comparatively weak effects of left-corner node counts might indicate that left-corner metrics are insufficiently predictive to account for the comprehension of head-final constructions of Dutch. The left corner of head-initial structures is very informative, which could explain why left-corner parsing metrics successfully predict brain activity of participants comprehending English [23,28]. We suggested in the introduction that these effects might be weaker in languages with head-final constructions, in particular if speakers of these languages adopt predictive parsing strategies. However, in seeming conflict with this possibility, a recent study in Japanese, a strictly head-final language, showed that a left-corner parsing model outperformed a top-down parsing model in left inferior frontal and temporal-parietal regions [109]. One relevant difference with our study is that they used a complexity metric that considers the number of possible syntactic analyses at each word (i.e., modeling ambiguity resolution), rather than directly quantifying the number of operations that are required to build the correct structure (i.e., node count for a one-path syntactic parse tree). These metrics do not reflect the same process. It is therefore possible that brain activity corresponding to ambiguity resolution is best modeled by considering the number of syntactic analyses following a left-corner strategy, and that, when the most likely analysis is chosen, the structure-building process itself is best modeled via top-down metrics. To better understand such diverging results, it is important that future studies take into account how syntactic properties (of different languages) affect the suitability of different parsing strategies. As a case in point, it is commonly mentioned that the left-corner strategy predicts processing breakdown for exactly those constructions that are difficult to process. Left-corner parsers have the property that their memory demands increase in proportion to the number of embeddings in center-embedded constructions, while they remain constant for both right- and left-branching structures [36–38]. Sentences with multiple levels of center-embedding indeed quickly over-tax working memory resources [31], supporting the cognitive plausibility of the left-corner method. Intriguingly, however, processing difficulty for center-embedded constructions is not consistent across languages. For example, it has been found that speakers of German (a language with head-final VPs, like Dutch) are hindered less than English speakers during the comprehension of multiply center-embedded sentences [43]. Language-dependent findings such as these reinforce the idea that people's ability to generate syntactic predictions might be dependent on the specific grammatical properties of the language. Because the average dependency lengths in head-final languages are longer than those in head-initial languages [110], speakers of head-final

languages have more experience with processing longer dependencies and might therefore rely more strongly on a processing strategy that facilitates the comprehension of these structures.

In all, the fact that node counts derived from a top-down parser best explain brain activity of people listening to Dutch stories might have to do with certain grammatical properties of Dutch, including its head-final VPs, which make left-corner prediction inadequate. That this conclusion can be reached merely by extending the approach to a closely related Germanic language underscores the need for more work on typologically diverse languages, whose structural properties invite different parsing strategies that might rely on different brain regions to varying degrees [44]. Overall, the fronto-temporal language network is remarkably consistent across speakers of different languages [111], but structural differences within this network can be induced by experience with sentence structures that elicit different processing behavior [112].

## 4.2. Is node count the right linking hypothesis?

Two critical questions can be raised about our use of incremental node count as the complexity metric to represent syntax-related neural states. First, the syntactic structures that we used to compute node counts contained empty elements, such as traces [25]. As these do not have an acoustic correlate in the physical stimulus, we assigned their node counts to the subsequent word (see Section 2.3). We reasoned that the existence and location of an element, whether covert or overt, can usually be inferred with absolute certainty only after it has been encountered, which would be at the subsequent word. However, this wait-and-see (or wait-and-infer) attitude is somewhat inconsistent with the parsing strategy of both top-down and left-corner parsers, which build structure predictively. On the top-down method, for instance, a constituent node is postulated before there is any evidence for its existence. Given that the structure corresponding to overt elements is built predictively, it is inconsistent if the structure corresponding to covert elements is built in an integratory manner, in particular given the evidence for prediction of null forms [87,89].

In order to check whether the node counts for traces were assigned correctly, the reconstruction accuracy of syntactic predictors must be tested when node counts for traces are assigned to the previous word. This will shift the structural complexity of the sentence to a different point in time and will do so differently depending on both the location of the trace and the parsing method. For traces at the left corner of a constituent (e.g., the first trace in Fig 2), whose node counts are higher for top-down than for bottom-up parsers, the preceding word will be assigned a higher node count on the top-down method. The reverse is the case for traces at the right corner of a constituent (e.g., the last trace in Fig 2), because their node counts are higher on the bottom-up method. Clearly, the method of assigning node counts of traces to the preceding or next word has important consequences, again showing that different ways of expressing syntax in time lead to different predictions about the dynamics of structure building.

Another relevant question is whether incremental node count is the right measure to represent syntactic structure building. The node count metric we used is unlexicalized, which means that it does not take into account the label of the node counted. During language comprehension, however, syntactic processing is lexicalized [33,94], and lexical information guides predictive structure building [39–41]. Such lexically driven structural predictions are represented to some extent in other metrics, such as surprisal values derived from probabilistic context-free grammars [18,20–21] or recurrent neural network grammars [15,17,109]. Both types of grammars incrementally build hierarchical structure, which they use to conditionalize the probability of an upcoming word or an upcoming word's part-of-speech.

There are at least 2 other metrics that are informative about syntactic processing in a way that incremental node count is not. First, the "distance" metric counts the total number of syntactic analyses that are considered by a parser at every individual word [15,17,109]. The larger the number of alternative analyses to be considered, the higher the effort in choosing the correct parse. In this way, it models the ambiguity resolution process that is much studied in psycholinguistics but that is not captured by node count. Second, "incremental memory" reflects the number of phrases to be held in memory on a stack [23,113] and is sometimes quantified as the number of open nodes [23,34]. Incremental memory is particularly relevant to evaluate left-corner parsing because one of the arguments in favor of the plausibility of the left-corner method relies on a complexity metric that reflects the number of unattached constituents to be held in working memory [36]. Node count instead quantifies syntactic complexity rather than memory load, and it need not be the case that these make exactly the same predictions with respect to comprehension difficulty.

## 5. Conclusions

In order for neuro-computational language models to be useful for the study of human biology, they must both be linguistically interpretable and be able to extend linguistic structure in time. The current study achieves both goals by using incremental node count as the linking hypothesis that connects atemporal syntactic structure with continuously varying neural dynamics. Node counts were derived from syntactic structures that were generated via an expressive and linguistically interpretable grammar. As such, syntax was extended into the time domain, yielding time-varying complexity metrics that could be regressed against delta-band neural activity in temporally resolved manner. By explicitly positing which parsing strategies are mostly likely to be entertained by the brain, this approach additionally yields predictions about the functional role of these brain regions. Results show that the top-down strategy, which is maximally eager and postulates syntactic structure early in time, best reflects the neural response in inferior frontal and superior temporal regions, suggesting that these regions are involved in building syntactic structure in a predictive manner. That being said, these results stay silent on the question what kinds of computations are implemented in these areas. This has to do with the fact that we computed node count based on derived tree structures rather than on the actual derivation trees. In order to build neuro-computational language models that are closer to cognitive and neurobiological processing, it is important that future work assumes a more transparent relation between grammar and parser, for instance by using the derivation steps directly implemented by the grammar [15,17,24,114]. Knowing when the brain projects this linguistic knowledge onto its perceptual analysis of speech is essential in order to understand how the brain transforms continuous sensory stimulation into cognitive representations, still a major question in human biology.

## Supporting information

**S1 Text. Results of additional control analyses.**
(PDF)

**S1 Fig. Correlation matrix for all word-based predictors.** The values correspond to the Pearson correlation for each pair of predictors. Data are available on the Radboud Data Repository (https://doi.org/10.34973/m1vp-hc15).
(TIFF)

**S2 Fig. Sources of (improved) explained variance of the base model and the syntactic models.** The results for the base model are presented in **(A)**. The results for the syntactic models

correspond to the predictors reflecting node count from the bottom-up (**B**), top-down (**C**), and left-corner (**D**) methods. All clusters that were significant at uncorrected alpha = 0.05 are displayed. Notice that the scales of the color bars are different across the plots. The plots on the right show the temporal response functions for node count derived from the bottom-up, top-down, and left-corner parsers in their respective models. Each line reflects the response function in a source point that was part of a cluster which showed a significant improvement in reconstruction accuracy. Data are available on the Radboud Data Repository (https://doi.org/10.34973/m1vp-hc15).
(TIFF)

**S3 Fig. Sources of the TRFs for node count derived from bottom-up, top-down, and left-corner parsers.** The colors represent t-values in source clusters that were significantly responsive (at corrected alpha = 0.0083) to the predictor in the indicated time windows. The midpoints of the color scales are at 70% of the maximum, such that the colored plots highlight the sources with the strongest effects. Data are available on the Radboud Data Repository (https://doi.org/10.34973/m1vp-hc15).
(TIFF)

**S4 Fig. Region of interest analysis. (A)** Spatial extensions of the 3 regions of interest. (**B**) Difference in reconstruction accuracy with the base model, plotted for left IFG, PTL, and ATL. The height of each bar indicates the improvement in reconstruction accuracy when only the relevant syntactic predictor was added to the base model. The drops represent the accuracy difference for individual participants, and the error bars represent the standard error of the mean across subjects. (**C**) Temporal response functions for node count derived from bottom-up, top-down, and left-corner parsers in their respective models. Error bars reflect the standard error of the mean per time sample. The horizontal bars below the TRFs reflect the temporal extensions of the significant clusters from the pairwise analysis. Data are available on the Radboud Data Repository (https://doi.org/10.34973/m1vp-hc15).
(TIFF)

**S5 Fig. Additional analysis to control for the correlation between structural and statistical predictors.** We repeated our analysis with TRF models from which the statistical predictors word frequency, surprisal and entropy were omitted. Results are shown separately for the effects of the bottom-up (**A**), top-down (**B**), and left-corner (**C**) predictors, and reflect the sources of significantly improved explained variance and temporal response functions in significant source points. Significance was determined by comparing the reconstruction accuracy of the full model to the reconstruction accuracy of a null model from which the relevant predictor was omitted. All clusters that were significant at uncorrected alpha = 0.05 are displayed. Notice that the scales of the color bars are different across the source plots. As explained in S1 Text, section 2, this pattern of results is similar to the results in Fig 4 of the main manuscript, showing that the relative effects of the syntactic predictors are stable when word frequency, entropy, and surprisal are omitted. Data are available on the Radboud Data Repository (https://doi.org/10.34973/m1vp-hc15).
(TIFF)

**S6 Fig. Additional analysis to control for the effect of prosodic boundary strength.** We repeated our analysis with TRF models to which we added a prosodic predictor. The results reflect sources of significantly improved explained variance and temporal response functions in significant source points, shown separately for the effects of the syntactic predictors bottom-up (**A**), top-down (**B**), and left-corner (**C**), and the prosodic predictor boundary strength (**D**). Significance was determined by comparing the reconstruction accuracy of the full model

to the reconstruction accuracy of a null model from which the relevant predictor was omitted. All clusters that were significant at uncorrected alpha = 0.05 are displayed. Notice that the scales of the color bars are different across the source plots. As explained in S1 Text, section 3, this analysis shows that the top-down effect is stable and independent of prosody, and that the variance explained by the bottom-up and left-corner predictors is reduced by the addition of prosodic boundary strength as predictor. Data are available on the Radboud Data Repository (https://doi.org/10.34973/m1vp-hc15).
(TIFF)

**S7 Fig. The effect of predictability on integratory structure building.** We included an additional analysis in which the effect of integratory structure building was evaluated separately for low- vs. high-surprisal words. **(A)** Significant sources of the accuracy differences between bottom-up node counts for low- vs. high-surprisal words. The positive accuracy difference indicates that the reconstruction accuracy was higher for the model with low- than for the model with high-surprisal words. **(B)** Temporal response functions for the bottom-up predictor, estimated separately for high-surprisal and low-surprisal words. Each line reflects the TRF estimated for a source point that was significant in the reconstruction accuracy analysis (i.e., the colored areas in (A)). **(C)** Amplitude of the response functions of the bottom-up predictor for low- and high-surprisal words, averaged over significant sources. The error bars reflect the standard error of the mean per time sample. The horizontal bar below the TRFs reflects the temporal extension of the largest cluster indicating the significant difference between the TRF amplitudes. Data are available on the Radboud Data Repository (https://doi.org/10.34973/m1vp-hc15).
(TIFF)

**S1 Table. Auditory stimuli.**
(PDF)

**S2 Table. Predictors included in each model.**
(PDF)

## Acknowledgments

We thank Laura Giglio for feedback on a previous version of this manuscript; Sophie Slaats and Michelle Suijkerbuijk for help creating the syntactic annotations; Noémie te Rietmolen for making Fig 1; and Ryan Law, Ioanna Zioga, Hugo Weissbart, and Sophie Slaats for contributing to data acquisition.

## Author Contributions

**Conceptualization:** Cas W. Coopmans, Helen de Hoop, Peter Hagoort, Andrea E. Martin.

**Data curation:** Cas W. Coopmans.

**Formal analysis:** Cas W. Coopmans, Filiz Tezcan.

**Investigation:** Cas W. Coopmans.

**Methodology:** Cas W. Coopmans.

**Project administration:** Cas W. Coopmans, Andrea E. Martin.

**Resources:** Cas W. Coopmans.

**Software:** Cas W. Coopmans, Filiz Tezcan.

**Supervision:** Helen de Hoop, Peter Hagoort, Andrea E. Martin.

**Validation:** Cas W. Coopmans, Andrea E. Martin.

**Visualization:** Cas W. Coopmans.

**Writing – original draft:** Cas W. Coopmans.

**Writing – review & editing:** Cas W. Coopmans, Helen de Hoop, Filiz Tezcan, Peter Hagoort, Andrea E. Martin.

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
