## [Editor Report · Decision Letter 0]

18 Mar 2024

Dear Dr Coopmans, 

Thank you for submitting your manuscript entitled "Neural dynamics express syntax in the time domain during natural story listening" for consideration as a Research Article by PLOS Biology.

Your manuscript has now been evaluated by the PLOS Biology editorial staff and I am writing to let you know that we would like to send your submission out for external peer review. Please note that we haven't been able to discuss your study with an Academic Editor so far. Therefore, we will make a call on the conceptual advance when we discuss the reviewer reports with the Academic Editor after review.

Once your full submission is complete, your paper will undergo a series of checks in preparation for peer review. After your manuscript has passed the checks it will be sent out for review. To provide the metadata for your submission, please Login to Editorial Manager (https://www.editorialmanager.com/pbiology) within two working days, i.e. by Mar 20 2024 11:59PM.

Kind regards,

Christian

Christian Schnell, PhD

Senior Editor

PLOS Biology

cschnell@plos.org

---

## [Decision Letter · Decision Letter 1]

31 May 2024

Dear Dr Coopmans,

Thank you for your patience while your manuscript "Neural dynamics express syntax in the time domain during natural story listening" was peer-reviewed at PLOS Biology. I'm handling your paper temporarily while my colleague Dr Christian Schnell is out of the office. It has now been evaluated by the PLOS Biology editors, an Academic Editor with relevant expertise, and by three independent reviewers. 

You'll see that reviewers #1 and #2 are quite positive and mainly request further methodological details and a couple of additional analyses. By contrast, reviewer #3 is more critical in a way that may reflect underlying differences in viewpoint, but also identifies many strengths. After cross-commenting between the reviewers, and further discussion with the Academic Editor, we would like to invite you to revise the work to thoroughly address the reviewers' reports.

Given the extent of revision needed, we cannot make a decision about publication until we have seen the revised manuscript and your response to the reviewers' comments. Your revised manuscript is likely to be sent for further evaluation by all or a subset of the reviewers.

**IMPORTANT - SUBMITTING YOUR REVISION**

*Re-submission Checklist*

*Published Peer Review*

*PLOS Data Policy*

*Blot and Gel Data Policy*

Sincerely,

Roli Roberts

Roland G Roberts PhD

Senior Editor

PLOS Biology

rroberts@plos.org

on behalf of

Christian Schnell, PhD, 

Senior Editor

PLOS Biology

cschnell@plos.org

REVIEWERS' COMMENTS:

Reviewer #1:

SUMMARY

The authors report an n = 24 MEG study on the neural correlates of syntactic structure building in the human brain. Using a modeling approach comparing different parsing algorithms, they relate MEG activity in specific time windows in specific brain areas to specific cognitive operations assumed to be performed during syntactic structure building, with the psychological goal of arriving at an interpretable representation of what we call sentences.

The analyses are done very adequately. The authors rightly go all the way from a representational assumption via a parsing algorithm and complexity metric to a state-of-the-art regression model on brain data of high spatio-temporal resolution. The authors' writing shows they know the psycholinguistic reasoning behind the parsing algorithms they are employing, so the whole path from cognitive science to the highest-possible-with-extracranial spatiotemporal resolution is taken. I enjoyed reviewing this manuscript, because this is how I know this type of work must be performed. The standard is very high here and it is an example for the readership who wishes to be inspired.

I list multiple points for further improvement of this impressive work below.

MAJOR

(1) Distribution of age: The age range measured here is rather wide. I think that more efforts need to be taken to estimate the generalizability of the current results and whether they hold beyond young / middle-aged / senior adults. Specific comment:

245: The age range is non-standard. There is evidence that syntactic processing changes quite a bit already around 50, if not earlier. Please comment on this / address by supplementary analysis. I would also like to know how age was distributed across the sample to judge whether the current findings can be taken as representative for adult processing, or whether the findings could even be restricted to the aging brain. In particular with respect to the idea that top-down processes are stronger in the aging brain + the shallow parsing concept / its relationship to age, this could be an interesting additional research question to be addressed, e.g., by correlations with an age predictor. Please comment / consider.

(2) Prosody / boundaries: What is the covariance of node count and prosodic boundary strength? Strong enough to pose a confound? Control analyses should be performed. Specific comment:

351ff: Was there a statistical relationship between the strength of prosodic boundaries and node count? Please provide supplementary analysis on this (e.g., Inbar et al., MEG, 2023,'s recently employed method).

(3) Refining the search space for time course analysis: Why not mask the time course analysis spatially with the clusters shown in Figure 4 to ditch new areas popping up that are hard to interpret? The authors use the timing of the TRF accuracy for masking, but (if I understand correctly) not the space? Specific comment:

527: The source configurations in this figure differ drastically from those in the previous figure. E.g., "Top-down": How can it be that the top-down parser improves TRF accuracy in the left hemisphere (and a bit right aTL), but then the time courses are all strong in the right hemisphere, too? Why not mask the source analysis of the time courses (Figure 5) with the results / maps shown in Figure 4? This would get rid of areas where the TRF accuracy is not improving. 

(4) Relating the findings to the structural literature: We do need to link the timing of syntactic processing to the underlying structural neuroanatomy, and the current findings allow for doing so (at least a bit, in the form a discussion).

575ff: The authors should make an effort to relate their results to the structural neuroanatomy of syntax (dorsal & ventral networks, prior proposals on what each network does & how these proposals relate to the cognitive operations modeled in the current submission). Linking the MEG time course world to the structural architecture of syntax is desperately needed in the literature, and the current time courses & source configurations (in particular when making use of my point about masking the Figure-5 analyses with the Figure-4 results, spatially) would allow for some rather solid discussion (I encourage the authors!)

(5) How do the current results relate to the classical ERP/F literature (e.g., ELAN / sMMN)? Can the authors discuss the prediction / integration debate? How do the current results relate to source localizations of these ERPs/Fs and / or the MRI literature using similar paradigms? It is fine to discuss the Matchin review, but it would increase your readership of you would also speak to the psycholinguistics concepts / issues a bit more (in the discussion).

MINOR

1-20: "yet, syntactic rules are" pls. change to *are widely assumed to be* or *many theories of human language* or something else, as this is not an established fact. No one really knows whether and how and where syntactic representations exist in the physical & cognitive sense, in particular whether there are atemporal representations (e.g., a whole tree of a whole sentence being present in the mind/brain in a given instant).

28: "words hierarchically are" pls. change this to be non-factual. The authors should write that *leading theories in theoretical linguistics assume that words are...* and so on. Again, these are not facts, but hypotheses / theories.

29f: "are abstract and not bound" see previous comments. These are hypotheses / theories.

32: Introduce term "parsing" to the non-experts, e.g., "The incremental buildup of syntactic structure over time is termed parsing in psycholinguistics".

33: "dynamics" -> "temporal dynamics" or "word-by-word dynamics"

36: I am not sure that "linguistic theory" is at the "computational level". The authors said that they assume that the initial input (production) and final output (comprehension) are an atemporal representation of syntactic structure. Now how is this computational? In my understanding, the buildup of this structure over time is computational, no?

43: Bever & Poeppel (2010) should be added to the references here I believe.

54: "has to be" -> "is assumed to be built when presupposing that the input / output [...] a hierarchical structure built from [...]"

74: I am not sure whether a complexity metric quantifies a "states". To me, a state of sth. qualitative (as in: one state compared to another state). Can the authors say rather that the metric quantifies the magnitude of a state / how strong a given state is assumed to be by the hypothesis?

76: What is meant by "relevance"? Please unpack.

79: What is "linguistic competence"? Do the authors mean "acquired knowledge of syntactic structure / rules"?

86f: This is unclear to me. What are "natural language structure" and, in particular, where are these (mind/brain/textbook)? I understand the "performance" bit better (~ words are incrementally integrated into some kind of syntactic structure on the fly), but what does this have to do with some kind of "competence" / how would this "competence" be different, physiologically / cognitively? If the authors think this is important, they should (a) unpack their reasoning a little more and (b) explain to the non-experts a bit about the historical genesis of the dualism of competence / performance and why it is relevant, in their view, to understanding the neural implementation of syntactic processing.

94: Motivate use of one particular linguistic theory, pls. Why is this theory a better theory than others / why did the authors choose it instead of other options?

96: "natural language structures" Please explain to the non-expert reader why "movement" is such a structure and possibly also what movement is.

166: Unpack for non-experts please—what is a complement?

237: Somewhat fuzzy. Effects in the literature start at ~80 ms (Herrmann et al.'s MEG, ca. 2009) and continue all the way until the P6 / CPS time windows (i.e., beyond 500 ms, considering reanalysis / model update / syntactic prediction error to be part of "structure building")... Maybe reformulate a bit to mark that your hypotheses are for the immediate word-by-word integrative / predictive parts (*first stage*)?

252f: Syntax in these stories, some > 150 years old, is certainly different from current syntax of Dutch. The authors should comment on this / address this. That is, not all structure in these stories can be taken to be "naturally occurring".

377f: It is a bit unclear to me how word-level complexity metrics are derived from GPT. Depending on the tokenizer, the probabilities delivered by GPT are not for each word, but for each sub-word token. Please comment.

408: I am not sure why such a long time window is chosen. (1) The window includes a substantial number of next-words, (2) the complexity metrics are already contextual, so the TRF calculations and the metrics may interact in some way the longer the deconvolution time window is? Please consider / comment.

704: "ability [...] dependent on the [...] grammatical properties" Please unpack. It is rather that (a) languages have certain grammatical properties, (b) these properties pose certain cognitive tasks, (c) different languages have different properties, hence magnitudes of different cognitive tasks are different across languages, (d) people are trained on specific languages / grammatical properties, (e) people are skilled to process particular languages, (f) brain responses depend on experience of people with languages (e.g., people have learnt that in language X, making predictions helps processing, whereas in language Y, making predictions is not helpful.).

730: For the non-experts, the authors might want to explain what traces are; see my above point about movement.

Reviewer #2:

The manuscript "Neural dynamics express syntax in the time domain during natural story listening" describes an experiment in which participants were listening to unaltered, naturally read stories in Dutch while neural activity was recorded using MEG. The authors use regression analyses to isolate neural signal variance that can be attributed to syntactic processing. Since syntax is abstract and atemporal, they use a proxy measure, incremental node count, to translate from different models of syntactic structure building to neural responses in the time domain. The different models build the same syntactic structure but differ in when syntactic nodes are added to the tree, either predictively (top-down), adding nodes to the tree before all child nodes are received, or bottom-up, once all the child nodes are received (+ an intermediary 'left-corner'-strategy). Although all models build the same syntactic tree with the same number of nodes, the time at which nodes are added to the tree differs, thus the measure 'incremental node count' is high/low at different parts within a sentence.

This strategy (using a processing difficulty or load measure for studying syntactic structure building during natural speech) has been used in previous studies on this topic, as the manuscript acknowledges. The novelty of the current study, according to the authors, lies firstly in the more comprehensive inclusion of control predictors (acoustic and word-level predictors) and secondly in the finding that top-down syntactic structure building explains a larger part of the neural variance compared to the other models: a finding that is in contrast to similar studies with English material. The authors attribute this to a difference in word-order between English and Dutch which render predictive structure building a more efficient strategy in Dutch. 

As a reader whose research focus is not on syntax, I really appreciated the very clear introduction and layout of the research problem. Studying sentence processing using explicit and transparent models, as the current paper does, promises to be very insightful and deserves to be published to the attention of a broad audience, especially as an alternative to the currently prominent approaches using black-box algorithms like large language models. However, I see some problems with the current paper which would need to be addressed to see whether the main claims of the paper are supported.

1) Effect of control predictors on the model comparison

Since this study is using unaltered natural speech material, it has to deal with the problem that different linguistic predictors are correlated. The authors' strategy is to add control predictors to their regression model. This means that e.g. in Figure 4, the additional R2 for the bottom-up model represents the unique variance explained by the bottom-up model, over and above what can be explained by the combination of acoustic predictors, word-level predictors (frequency & GPT2 surprisal) and the other two syntactic models. This is done to strengthen the evidence that explicit syntactic models explain neural activity above and beyond competing models such as GPT2: 

Page 28, line 597: "the null models also contained information-theoretic predictors (e.g., surprisal, entropy) that partially reflect syntactic information and therefore explain some of the variance in brain activity whose origin is syntactic. By including these semi-syntactic control predictors, we stacked the cards against us and likely underestimated the 'true' neural response to syntactic structure"

I agree and it strengthens the basic finding that the neural activity described reflects syntactic structure building. However, the control predictors might bias the comparison between the three syntactic models due to correlations with the syntactic predictors. Looking at the correlation table in Figure S2.1, it seems that the top-down regressor is less correlated with the control regressors than bottom-up or left-corner regressor. So it could be that GPT2 surprisal is correlated more to bottom-up node count and thus decrease the neural variance explained by that model. The correlations with GPT2 entropy/surprisal are overall small, so this might be not be a big problem. But given that the unique neural variance explained is small (e.g. unique R2=5e-05 for the top-down model), it is possible that there is a bias. Does the top-down model still outperform the other models when GPT2-based regressors are removed? A change in effect would be quite problematic for the claims of the paper, especially because GPT2 is a black box model, so it would be quite difficult to explain theoretically.

The correlations are even stronger with the word frequency regressor: .33 for bottom-up, -.081 for top-down. Where does this come from? Looking at the example sentence in Figure 2: Under the top-down model, high node counts are assigned to the beginning of phrases which in this case are mostly function words (with high frequency) and under the bottom-up model, high node counts fall more on content words (with lower frequency); an example phrase being "een doordringende kreet". Is this a pattern that is representative of the corpus? Does this affect the main findings of the paper: namely is the higher R2 of the top-down model due to the inclusion of the control regressor word frequency? Do the other papers in the field use these control regressors, and can the divergent findings be due to that difference, rather than a difference in language Dutch vs. English? Without addressing these questions it's difficult to make comparisons to other papers, or claims about language-related differences.

2) Temporal response functions on the source level - polarity

The source maps of the TRFs (Figure 5) are quite difficult to digest, compared to the other results. Is it correct that you are estimating the t-statistics from the unrectified TRFs? The problem with MEG source maps is the spatial spread: e.g. the activity of a ground truth source in the superior temporal gyrus would spread to insula, the STS and the opposite side of the Sylvain fissure on the frontal/parietal operculum, or even further depending on how strong the activity of the ground truth source is. When an unsigned value is plotted such as the accuracy of a regression model (as in Figure 4) this is less problematic, because the activation then appears as one statistical cluster with a focus on the STG (Figure 4B). When a signed value is plotted such as a TRF, and the source model was computed with surface normal orientation constraint, discontinuities appear in the map: the same ground truth source would appear e.g. as a negative activation on the STG and as a positive activation on the opposite of the Sylvain fissure, because the dipoles modeled on those two sides of a sulcus have opposite orientations.

Do the authors think that all these positive and negative clusters in time represent distinct neural sources? If yes, they should be discussed and interpreted. In the current manuscript they are not. The interpretation of the time course of neural activity in the discussion section (page) comes from the time courses plotted in Figure 4 and 6, namely the early and late peak of the top-down model, and the peak of the bottom-up model in-between. This can be nicely seen in Figure 4. But Figure 5 is difficult to interpret. My suggestion would be to rectify the TRF time-series for plotting on the source maps (which is more common in MEG papers), this would potentially lead to larger more contiguous clusters, which can more easily be interpreted as one or a small number of sources + spatial spread. Or drop this figure if it is not interpreted.

A related question, which is not completely described in the method section: how are the ROI results generated? Do you create a new average time-series per ROI per subject and then compute accuracies, or just average accuracies over the ROI's source points? How are the TRFs per ROI computed? Are the issues with polarity that I described above problematic here? There could be cancellations when averaging over ROIs, which could mask results. Rectifying TRFs could also be helpful here.

Reviewer #3:

This paper has a very promising start, but it leaves the reader with an unclear outlook.

Major critical comments:

1. A first major point is that I am not sure whether the authors' goal is to uncover syntactic representations as they are deployed in time, or the processing issue of varying syntactic complexity. The metric is unclear to this end, even as originally proposed by Brennan. Therefore, it is unclear, at least to me, which cognitive processes the dependent measure taps into. I realize that the three parsing models tested are models of processing, but is the neural dependent measure a correlate of syntactic structure deployment (states), or of the incremental processes feeding into each state (changes across states)? A mixture of the two? It should be clarified.

2. A second major point is that syntactic information rate seems to be based on the assumption that not only syntactic complexity changes in time (numerator), but that the time range at the denominators is very much constrained, to a point that it makes sense to attribute any significant change in rate not to variability in the denominator, but rather to variability in the numerator. In the case at hand, this would mean that the rate of the specific medium - speech acoustics - is roughly constant, and what varies is the number of nosed closed (or opened) at each point in time. Such assumption does not stand on empirical grounds, but may have been inferentially derived from massive averaging approaches to speech acoustic analysis, which output a shallow energy peak in the low-theta, high-delta ranges. Those analyses are useful to approximate a range of interest, but the speech signal is much more dynamic on a local scale. And here is the key point: what counts as local scale in this paper? The authors use a very peculiar way of selecting relevant syntactic information (see point 6 below), sparsely extracting macroconstituents from the syntactic flow of information, a highly debatable choice which seems to artificially determine a roughly stable time variable in the denominator. 

3. A third major point is that the so-called "results linking syntactic processing to delta-band activity" is actually a set of theses or at best a mixed bag of results, including debated interpretations of frequency tagging findings and approaches using narrowband mutual information, the latter requiring aggressive low-pass filtering which almost invariably leads to false positives (MI is "created" by phase distortions due to LP filtering, rather than computed from the actual data). Hence, the motivation for the focus in the delta band does not have solid grounds. Which in turn greatly limits the outlook as well as the outcome of the paper. 

4. The variance in sentence duration (in number of words) is very high: lines 254-255, "791 sentences, which are on average 10.8 words long (range = 1-35, SD = 5.95)". That reflects variance in phrasal structure duration, so I am very doubtful that one single rate of syntactic processing can cover such variance, and I am even more convinced that it is an artefact of the macroconstituent selection approach.

5. I can understand the decision to leave out APs, but leaving out AdvP does not make much sense, given that they invariable include one or more, nested, NPs.

6. The section at lines 300-306 reflects the key problem with this paper, in my opinion, that is the selection of what is relevant data for the determination of a syntactic information rate. First, did the authors count also lexical nodes or not? In other words, what is node = 1? Only by knowing that can one understand exactly what node = 2 really means as a threshold. Second, selecting such a threshold is an arbitrary choice, bordering cherry picking (that is, selecting a data frame that yields results consistent with the unsubstantiated delta hypothesis). Are the authors only selecting macroconstituents for their syntactic analysis? And why would that be a meaningful way to go for a putative comprehension process? Do they assume that macroconstituents are processed in isolation by the rest of syntactic constituents by a separate sub-band of the delta band? I have a hard time with this choice of data analysis, honestly. 

7. The Results section sees the top-down model largely outperforming the remaining two models. Given that only macroconstituents can enter the computation, is it really that surprising a finding? Can conclusions about syntax in general (see the Title and Discussion) be derived from arbitrarily selecting only some syntax of choice? At the very least, the authors should test the models on the whole syntactic flow of information. 

8. Figure 6 is interesting but leaves the reader with the impression that a line of work could stem from there, but has been somewhat top-down interrupted.

Minor critical comments:

1. Line 112: The reduction operation cannot be identical to the expansion operation (of which at line 107): VP > V NP. Please, revise.

2. Line 345: "We low-pass filtered the signal at 4 Hz using a zero-phase FIR filter (corresponding to the 0.5-4 Hz delta band)". Does this mean that the transition band is 1 Hz for both the 0.5 HP and the 4 Hz LP? If that is so, then the band that is actually untouched is 1-3.5 Hz. I would suggest considering either a longer filter with a smaller TB or LP at 4.5 Hz. The HP seems to be difficult to change, unless the authors consider a longer filter. MNE procedures generally do not pay sufficient attention to filtering issues.

Major critical strengths:

1. I very much like the strategic use of the Dutch language mix headedness feature to test the validity of parsing strategies outside of the English language. I think this is the strong point of the paper. Had the authors then worked on the whole syntactic information flow rather than arbitrarily selecting some data to fit their hypothesis, it would have made for a nice paper. 

2. I appreciate a lot Figure 3. I do not share the method that led to the figure, but I appreciate the clarity and thoroughness of the illustration.

Minor critical strengths:

1. I appreciate the use of the "Ffmpeg software" instead of RMS for loudness normalization. However, was is the loudness level chosen to be equal for all participants or based on an individually measured threshold? 

In conclusion, I appreciate the technical effort, and commend the authors on the good quality of the illustrations, but I cannot get on board with the data selection, as it seems not to be data-driven, or even hypothesis-driven, as it is very hard to hypothesize that somewhat complex syntax lives in a frequency of its own. I am sorry I cannot be more positive.

---

## [Decision Letter · Decision Letter 2]

13 Nov 2024

Dear Dr Coopmans,

Thank you for your patience while we considered your revised manuscript "Neural dynamics express syntax in the time domain during natural story listening" for publication as a Research Article at PLOS Biology. This revised version of your manuscript has been evaluated by the PLOS Biology editors, the Academic Editor and the original reviewers.

Based on the reviews, we are likely to accept this manuscript for publication, provided you satisfactorily address the following data and other policy-related requests.

* We found the current title a bit inaccessible for our broad readership, but struggled to come with something better. Would "The neural dynamics that map the temporal dynamics of speech to syntactic structure during natural story listening vary by language" work for you?

* Please add the links to the funding agencies in the Financial Disclosure statement in the manuscript details.

* Please include information in the Methods section whether the study has been conducted according to the principles expressed in the Declaration of Helsinki.

* DATA POLICY:

Regardless of the method selected, please ensure that you provide the individual numerical values that underlie the summary data displayed in the following figure panels as they are essential for readers to assess your analysis and to reproduce it: 2, 6B, and S2.4B.

* CODE POLICY

* Please move the references and supplementary methos to the main manuscript. The supplementary figures can remain in that file. As we have no limitations in word count or number of references, methods should be described completely in the main manuscript. 

We expect to receive your revised manuscript within two weeks. 

*Published Peer Review History*

*Press*

Sincerely,

Christian

Christian Schnell, PhD

Senior Editor

cschnell@plos.org

PLOS Biology

Reviewer remarks:

Reviewer #1: Good job, authors! Thank you for the constructive and appreciative interaction! This work is highly comprehensive and balanced at the same time now!

Reviewer #2: I think the authors responded thoroughly to all my comments and the concerns have been addressed. The main result (node counts from top-down parser better explains neural data) seems robust to several control analyses that were suggested by me and other reviewers. I'm also particularly happy to see that the spatial distribution of results is more consistent now across different analyses, after the authors switched to working with rectified data.

Reviewer #3: I have read the revised paper more than once. I have read the authors' replies to my comments, and while the whole "syntax in the delta band" hypotheses remains largely unsubstantiated (the papers the authors suggest as support are far from supportive, in fact!), I am now convinced that the revised paper has reached the quality standards for publication in Plos Biology.

---

## [Editor Report · Decision Letter 3]

5 Dec 2024

Dear Cas,

Thank you for the submission of your revised Research Article "Language-specific neural dynamics extend syntax into the time domain" for publication in PLOS Biology. On behalf of my colleagues and the Academic Editor, Lars Meyer, I am pleased to say that we can in principle accept your manuscript for publication, provided you address any remaining formatting and reporting issues. These will be detailed in an email you should receive within 2-3 business days from our colleagues in the journal operations team; no action is required from you until then.

When you attend to these requests, could you please also specify in a bit more detail where exactly the source data can be found? I think most readers won't be able to find the source data with the current description.

Please note that we will not be able to formally accept your manuscript and schedule it for publication until you have completed any requested changes.

PRESS

Sincerely, 

Christian

Christian Schnell, PhD

Senior Editor

PLOS Biology

cschnell@plos.org